# 🧩 Assembling the Mind's Mosaic: Towards EEG Semantic Intent Decoding

**Jiahe Li[1]***, **Junru Chen[1]***, **Fanqi Shen[1]**, **Jialan Yang[1]**, **Jada Li[2]**, **Zhizhang Yuan[1]**,
**Baowen Cheng[3]**, **Meng Li[3,4,5]**, **Yang Yang[1]†**

[1]Zhejiang University, [2]University of Texas at Austin, [3]Shanghai Institute of Microsystem and Information Technology, Chinese Academy of Sciences

{jiaheli,jrchen_cali,shenfanqi,jlyang,zhizhangyuan,yangya}@zju.edu.cn
jada.li@utexas.edu, chengbaowen23@mails.ucas.ac.cn
limeng.braindecoder@gmail.com

## Abstract

Enabling natural communication through brain–computer interfaces (BCIs) remains one of the most profound challenges in neuroscience and neurotechnology. While existing frameworks offer partial solutions, they are constrained by oversimplified semantic representations and a lack of interpretability. To overcome these limitations, we introduce **Semantic Intent Decoding (SID)**, a novel framework that translates neural activity into natural language by modeling meaning as a flexible set of compositional semantic units. SID is built on three core principles: semantic compositionality, continuity and expandability of semantic space, and fidelity in reconstruction. We present **BrainMosaic**, a deep learning architecture implementing SID. **BrainMosaic** decodes multiple semantic units from EEG/SEEG signals using set matching and then reconstructs coherent sentences through semantic-guided reconstruction. This approach moves beyond traditional pipelines that rely on fixed-class classification or unconstrained generation, enabling a more interpretable and expressive communication paradigm. Extensive experiments on multilingual EEG and clinical SEEG datasets demonstrate that SID and **BrainMosaic** offer substantial advantages over existing frameworks, paving the way for natural and effective BCI-mediated communication.

## 1 Introduction

Despite advances in neuroscience and neurotechnology, understanding and restoring communication remains one of the most profound challenges in brain–computer interface (BCI) research. Conditions such as aphasia and locked-in syndrome can sever an individual's ability to speak or write, isolating them from even the simplest forms of interaction. BCIs, recording neural activity via scalp or intracranial electroencephalography (EEG), offer a promising pathway to bypass these physical barriers by translating brain signals directly into language. Most existing approaches fall into one of two primary paradigms: **Speech Decoding** and **Concept Decoding** (see Table 1 for a comparison). Speech Decoding aims to reconstruct overt or imagined speech from motor-related cortical areas. While recent advances have improved performance, this paradigm remains fundamentally constrained by its reliance on motor regions, which represent only a limited subset of the brain's expansive language network. This focus overlooks the distributed nature of semantic processing and limits its cross-linguistic generalizability by depending on phoneme-level reconstruction (Dronkers et al., 2017). In contrast, Concept Decoding seeks to directly extract the intended meaning of an utterance from neural activity (Zhang et al., 2024). Leveraging distributed neural activity, it supports communication for individuals with impairments and advances understanding of language and thought mechanisms. This paradigm thus extends BCI research towards decoding abstract cognition and paves the way for developing next-generation neurotechnologies.

---

*Equal contribution. † Corresponding author
Additional Affiliations: [4]Guangdong-Hong Kong-Macao Greater Bay Area Center for Brain Science and Brain-Inspired Intelligence, Southern Medical University, [5]INSIDE Institute for NeuroAI

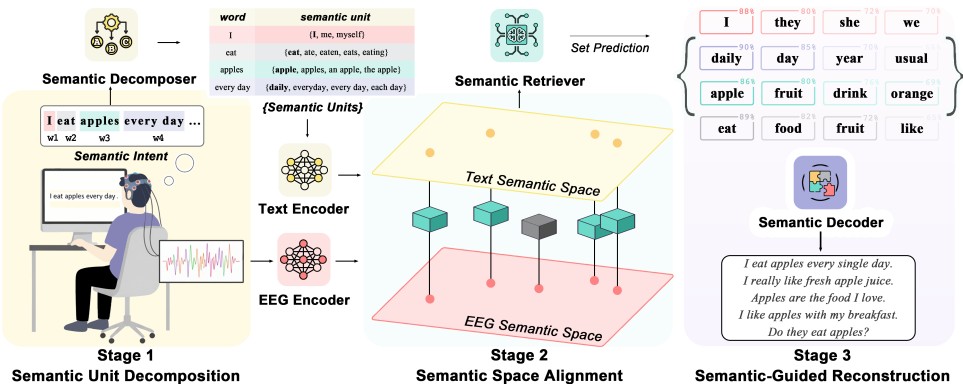

Figure 1: Overview of the Semantic Intent Decoding (SID) framework.

However, current implementations of Concept Decoding follow two contrasting frameworks, both presenting notable limitations. The first formulates concept decoding as a classification task on a fixed, predefined set of concepts or topics (Liu et al., 2009; Zhang et al., 2024). Though simple to implement, this approach is fundamentally restrictive. Discrete labels inherently struggle to capture the continuous and overlapping nature of meaning, often leading to poor alignment with real-world communicative intent. Alternatively, a more recent direction seeks to enhance expressive capacity by mapping neural signals directly into the latent representation space of large language models (LLMs) (Shams et al., 2025; Lu et al., 2025; Duan et al., 2023). However, these methods typically require vast amounts of paired neural-linguistic data and function as black boxes. Consequently, they lack interpretability and scientific transparency, making their output difficult to verify and control.

To address the limitations of existing frameworks in Concept Decoding, particularly the trade-off between interpretability and expressiveness, we introduce a novel framework: **Semantic Intent Decoding (SID)**. At its core is **Semantic Intent**, defined as a coherent unit of meaning that can be articulated in natural language. This notion spans a broad spectrum of communicative content, from casual, everyday expressions to formal and literary language. Table 2 summarizes representative datasets commonly used in Concept Decoding research and presents examples from each, illustrating the stylistic diversity encompassed by the definition of Semantic Intent. At a representational level, a Semantic Intent is not a single point in space but a flexible set of core semantic units. For example, the intent behind "I eat apples every day" can be represented as the set {I, eat, apple, daily}.

To translate this abstract representation into a practical decoding system, we design a three-stage pipeline (Figure 1). The SID framework comprises: (1) *Semantic Unit Decomposition*: The Semantic Decomposer transforms raw neural signals into a set of core semantic components; (2) *Semantic Space Alignment*: The Semantic Retriever matches neural representations within an open space of conceptual representations; and (3) *Semantic-Guided Reconstruction*: The Semantic Decoder assembles the retrieved elements into a coherent natural language utterance. Each stage is designed based on converging evidence from linguistic theory and cognitive neuroscience. Specifically, these three processing steps correspond to the three foundational principles outlined below, which reflect how meaning is internally represented, externally expressed, and inferentially reconstructed.

**1. Compositionality: A Semantic Intent can be naturally represented as a variable set of semantic units.** Human communicative intent is inherently multidimensional, comprising a finite, variable set of distinct yet interrelated units drawn from a vast conceptual universe. Consider the intent of "I eat apples every day", which decomposes into [I], [eat], [apple], and [daily]. Conversely,

Table 1: Comparison of brain-signal-to-language decoding paradigms and frameworks.

| Paradigm | Framework | Compositionality | Continuity | Expandability | Fidelity |
|---|---|---|---|---|---|
| **Concept Decoding** | SID (proposed) | Semantic Units | ✓ | ✓ | High |
| | Label Classification | Single Category | ✗ | ✗ | Low |
| | End-to-End Generation | Full Sentence | ✓ | ✓ | Low |
| **Speech Decoding** | Phonetic Reconstruction | Phoneme Sequence | N/A | ✓ | High |
| | Spectrum Mapping | Spectrogram | N/A | ✓ | Low |

Table 2: Overview of datasets in multiple languages for EEG semantic intent decoding, with illustrative examples of sentence styles.

| Dataset | Language | Style | Example |
|---------|----------|-------|---------|
| Chisco (Zhang et al., 2024) | Chinese | Daily | 今天是我的生日。
(Today is my birthday.) |
| ChineseEEG2 (Chen et al., 2025) | Chinese | Literary | 满地都是狰狞的石头。
(The earth bristled with jagged stones.) |
| ZuCo (Hollenstein et al., 2018) | English | Formal | As a remake, it's a pale imitation. |
| Clinical | Chinese | Daily | 我困了，准备睡觉了。
(I'm sleepy and ready for bed.) |

the sentence "I eat apples every day, usually the sweet ones my grandma picks during harvest" integrates attributes ([sweet]), source ([grandma picks]), and temporal context ([during harvest]). This complexity highlights why single-label classification cannot capture the richness of communicative intent (Zhang et al., 2024). This compositional perspective aligns with evidence that the brain constructs meaning through dynamic, context-sensitive processes rather than fixed, localized representations (Kutas & Federmeier, 2000; Zhang & Pylkkänen, 2018).

**2. Continuity and Expandability: Semantic units should be decoded within an open and continuous space.** Linguistically, semantic space is inherently open-ended and continuous: when new terms (e.g., Apple's "Vision Pro") emerge, speakers rapidly infer meaning from context and integrate them with related concepts (AR headsets, spatial computing). Thus, decoding should operate within an open, continuous space that supports similarity-based retrieval and seamless generalization. Neuroscientifically, the brain encodes concepts as smooth, low-dimensional maps across the cortex (Huth et al., 2012), paralleling behavioral similarity judgments and vector spaces in language models (Hebart et al., 2020; Doerig et al., 2025; Grand et al., 2022). Decoding into this continuous space enables generalization to novel concepts beyond fixed label sets.

**3. Fidelity: Reconstructed sentences must reflect decoded meaning and natural language structure.** From a linguistic perspective, fidelity has two dimensions. Semantic fidelity ensures that decoded units constrain generation, keeping outputs grounded in intended concepts (e.g., "apple," "harvest," "eat") rather than drifting to unrelated ones like "rabbit." Linguistic coherence ensures these units form grammatical, plausible sentences, avoiding violations like "apples eat the harvest." Together, these constraints guide contextually appropriate inferences such as "orchard" over "classroom," yielding faithful and intelligible outputs. From a neuroscience perspective, comprehension similarly depends on semantic congruity and structural control, supported by the brain's semantic control network for flexible, context-sensitive retrieval (Jefferies, 2013). Enforcing these constraints yields simultaneously interpretable and intelligible interpretations, circumventing the idiosyncratic errors common in unconstrained end-to-end decoding (Duan et al., 2023).

Guided by these principles, we introduce BRAINMOSAIC, a novel deep learning method for open-vocabulary communication within the Semantic Intent Decoding (SID) framework. BRAINMOSAIC decodes Semantic Intent through three stages: it first decomposes neural meaning by modeling EEG signals at the level of multiple semantic units; second, it aligns the decoded semantic space with a large-scale, open, and continuous linguistic space through set matching, aligning brain-derived units with linguistic concepts; and finally, it guides sentence reconstruction, leveraging the rich semantic representations and reasoning capabilities of large language models (LLMs) to generate coherent, faithful outputs. BRAINMOSAIC fully realizes the advantages of the SID framework, overcoming the limitations of existing mainstream approaches. The key contributions of this work are as follows:

1. We propose Semantic Intent Decoding (SID), a framework for EEG-based concept decoding that maps neural signals into a continuous semantic space, providing an interpretable link between concept-level representations and natural language generation.

2. To implement SID, we introduce BRAINMOSAIC, a concrete implementation of SID that predicts a variable-sized set of semantic representations via set matching and uses semantic-guided language modeling for constrained sentence generation.

3. Extensive experiments on both public multilingual EEG datasets and a private clinical Stereo-EEG (SEEG) dataset demonstrate that SID and BRAINMOSAIC deliver substantial improvements over existing frameworks.

## 2 METHOD

This section presents BRAINMOSAIC within the context of the SID framework, describing its architecture and the way each component implements the core principles of SID. We first outline the three foundational principles and key components of SID, explaining how BRAINMOSAIC is designed to address them. We then describe how these components are integrated into a unified system for semantic intent decoding.

### 2.1 SEMANTIC DECOMPOSER: DECOMPOSING NEURAL SIGNALS INTO SEMANTIC UNITS

**Principle 1** (Compositionality: Set-based Representation of Intent). A Semantic Intent $I$ can be represented as a variable set of semantic units $S = \{u_1, u_2, \ldots, u_n\}$. Semantic unit sets follow the fundamental properties of sets: they are unordered, contain no duplicates, and can vary in size.

A Semantic Intent corresponds to a single, coherent meaning expressed as a finite set of semantic units. While fixed word order is required for well-formed sentences, the core meaning of short, simple utterances can often be preserved despite changes in word order. Psycholinguistic evidence supports this approximation: eye-tracking studies show that reading is not strictly serial, with readers prioritizing semantic gist over exact token order (Schotter & Dillon, 2025), and transposed-word experiments in both English and Chinese demonstrate that moderate word swaps do not necessarily disrupt comprehension (Mirault et al., 2018; Liu et al., 2022). Moreover, the size of a Semantic Intent is bounded by cognitive constraints on working memory, which can only actively maintain a limited number of informational "chunks" at once (Cowan, 2001). This means a simple intent may contain only a few units (e.g., "I eat apples every day."), while a more detailed one may involve more (e.g., "I eat apples every day, usually the sweet ones my grandma picks during harvest."). Longer and more complex sentences can thus be naturally decomposed into multiple semantic intents, each forming a manageable, approximately permutation-invariant unit for modeling.

**BRAINMOSAIC Semantic Decomposer.** According to Principle 1, BRAINMOSAIC decodes EEG features into a fixed-size set of $K$ query slots, each representing a potential semantic unit. The value of $K$ is chosen based on the maximum expected number of semantic units for a single intent in the dataset, with the actual number of active units varying freely ($1 \leq n \leq K$). This design balances flexibility for sentence complexity and a clear upper bound. Inspired by set-based object detection frameworks such as DETR (Carion et al., 2020), we adopt a bipartite matching formulation to handle the variable and unordered nature of semantic units. During training, each ground-truth unit is uniquely matched to at most one predicted slot, while unmatched slots are assigned to a special "no-object" class. This converts the variable target set into a well-defined training signal, explicitly guiding the network to decompose EEG features into semantically meaningful components. The optimal matching and final training loss are defined as:

$$\hat{\sigma} = \arg\min_{\sigma \in \mathfrak{S}_K} \sum_{i=1}^{K} \mathcal{L}_{\text{match}}(y_i, \hat{y}_{\sigma(i)}), \quad (1) \qquad \mathcal{L}_{\text{Hungarian}}(y, \hat{y}) = \sum_{i=1}^{K} \mathcal{L}_{\text{match}}(y_i, \hat{y}_{\hat{\sigma}(i)}), \quad (2)$$

where $\mathcal{L}_{\text{match}}$ combines semantic alignment loss for matched slots and a binary classification loss for unmatched slots labeled as no-object. This ensures one-to-one matching and provides stable supervision regardless of the order and number of semantic units. In this way, BRAINMOSAIC achieves both permutation invariance and bounded cardinality as required by Principle 1, laying the groundwork for the next stage, where predicted units are aligned with a continuous semantic space for retrieval and reconstruction.

### 2.2 SEMANTIC RETRIEVER: ALIGNING SEMANTIC UNITS WITH CONTINUOUS SPACE

**Principle 2** (Continuity and Expandability: Continuous Representation of Semantic Space). Semantic units should be decoded within an open and continuous space $\mathcal{V} \subset \mathbb{R}^d$, where each unit $u$ is mapped to an embedding $E(u) \in \mathcal{V}$. In this space, similar concepts lie closer together, with semantic similarity inversely proportional to their embedding distance.

Semantic relationships are inherently continuous rather than discrete, as evidenced by findings from linguistics and psycholinguistics. From human cognition, the distributional hypothesis links contex-

tual co-occurrence to semantic relatedness (Harris, 1954), and phenomena such as semantic priming and graded synonymy show that similarity is gradual and quantifiable (Meyer & Schvaneveldt, 1971; Miller & Charles, 1991). The continuity of semantic space implies that human cognition open and compositional. New concepts emerge from the organic integration of existing ones into the global associative network. Consequently, semantic space does not expand indefinitely but instead becomes increasingly dense (Fodor, 1975; De Deyne et al., 2019). EEG studies further show that neural activity captures continuous semantic representations, with patterns in brain signals predicting graded word vector relationships beyond discrete category boundaries (Foster et al., 2021). Modern pre-trained LLM embedding spaces instantiate the same principle computationally (Zhang et al., 2025b; ByteDance, 2024). Trained with multi-stage contrastive objectives on massive corpora, they yield a stable manifold $\mathcal{V}$ in which vector proximity reliably tracks semantic affinity. Thus, the structure observed in human language and cognition can be mirrored in these learned spaces, providing a well-validated target for aligning EEG-derived semantic units to precise coordinates within $\mathcal{V}$.

**BRAINMOSAIC Semantic Retriever.** Building on Principle 2, the BRAINMOSAIC Semantic Retriever uses bipartite matching procedure to align each predicted slot with a ground-truth semantic unit in the continuous space $\mathcal{V}$. The per-slot matching loss, $\mathcal{L}_{\text{match}}$, consists of two terms: a semantic alignment term that encourages the predicted embedding $\hat{y}$ to be close to its corresponding target word embedding $E(u)$ based on cosine similarity, and a slot activity classification term that distinguishes active semantic units from "no-object" slots. This joint objective enables the model to learn token-level semantic alignment.

$$\mathcal{L}_{\text{match}}\big(E(u), \hat{y}, t, \hat{p}\big) = t\big[\,1 - \text{sim}\big(E(u), \hat{y}\big)\,\big] \;+\; \lambda_{\text{cls}}\Big(-t\log\hat{p} - (1-t)\log(1-\hat{p})\Big), \quad (3)$$

Here, $\text{sim}(\cdot, \cdot)$ denotes a cosine-based similarity, $t \in \{0, 1\}$ is the slot's match indicator (1 if the slot is assigned to a ground-truth unit; 0 for "no-object"), and $\hat{p}$ is the predicted activity probability. This ensures that only matched slots are pulled toward their targets in $\mathcal{V}$, while all slots learn a calibrated decision boundary between active and "no-object". In parallel, the retriever predicts a global embedding $\hat{s} \in \mathcal{V}$. This embedding is aligned with the ground-truth sentence embedding $E(s)$ to capture holistic semantic intent, and is further processed through multiple classification heads. Each head predicts a distinct global attribute of the intent (e.g., tone, subjectivity). This dual supervision enables the model to integrate token-level information with high-level global properties.

$$\mathcal{L}_{\text{global}}(E(s), \hat{s}, Z, \hat{Z}) = \underbrace{\big[\,1 - \text{sim}(E(s), \hat{s})\,\big]}_{\text{global alignment}} + \lambda_{\text{attr}} \underbrace{\sum_{c=1}^{C} \text{CE}\big(z^{(c)}, \hat{z}^{(c)}\big)}_{\text{global attribute classification}}, \quad (4)$$

where $z^{(c)}$ and $\hat{z}^{(c)}$ are the ground-truth and predicted outputs for the $c$-th global attribute, $\lambda_{\text{attr}}$ balances the alignment term and the classification term. The overall retriever loss combines the Hungarian objective from the token-level matching with the global-level supervision.

$$\mathcal{L}_{\text{retriever}} = \mathcal{L}_{\text{Hungarian}} + \lambda_{\text{global}} \cdot \mathcal{L}_{\text{global}}, \quad (5)$$

## 2.3 SEMANTIC DECODER: RECONSTRUCTING LANGUAGE FROM SEMANTIC UNITS

**Principle 3** (Fidelity: Faithful and Coherent Reconstruction). Given a predicted set of semantic units $S_{\text{retrieved}} = \{u_1', \ldots, u_m'\}$, the reconstruction process seeks a sentence $T = G(S_{\text{retrieved}})$ that is both semantically grounded in these units and linguistically well-formed.

Modern LLMs, trained on massive and diverse text corpora, exhibit exceptional capabilities in contextual reasoning, compositionality, and natural language generation (Achiam et al., 2023; Team et al., 2023; Liu et al., 2024). These properties make them particularly well-suited for reconstructing fluent, coherent sentences from a discrete set of semantic units.

**BRAINMOSAIC Semantic Decoder.** In line with Principle 3, the Semantic Decoder in BRAINMOSAIC is built around a LLM. To interface with it, Semantic Retriever outputs are first converted into a structured prompt. For each predicted slot, the retriever searches the entire semantic space and returns returns ranked candidate units with probabilities (reflecting importance). Collectively, this

matching process yields the final retrieved set $S_{\text{retrieved}} = \{u_1', \ldots, u_m'\}$ with corresponding probabilities $\{p_1, \ldots, p_m\}$. These token-level candidates are then integrated with global sentence-level attributes predicted by the model, such as sentence type or tone, to form the final prompt:

$$\text{Prompt} = P(S_{\text{retrieved}}, Z) \qquad T = G(\text{Prompt}), \tag{6}$$

where $Z$ denotes the set of global contextual signals. Through this, LLM $G$ reverses semantic decomposition to integrate retrieved units into a fluent sentence. Output $T$ remains faithful to decoded units and robust to retrieval noise.

### 2.4 FORMAL DEFINITION AND ASSEMBLY OF BRAINMOSAIC

Figure 1 provides an overview of the complete workflow, reflecting the overall structure of the Semantic Intent Decoding framework on which BRAINMOSAIC is grounded. Other components and the detailed algorithm description are provided in Appendix J.

During **training**, raw EEG recordings are first processed by the *EEG Encoder*, which converts multi-channel temporal signals into a set of neural feature tokens. These tokens are passed to the *Semantic Decomposer*, where the Transformer and its learnable queries produce $K$ candidate slots representing potential semantic units. The *Text Encoder* provides the reference semantic space, including unit-level embeddings and sentence-level targets. The *Semantic Retriever* aligns each predicted slot with this space through bipartite matching, simultaneously predicting token-level assignments and global intent attributes. The entire network is optimized end-to-end using the combined losses introduced above, ensuring that neural features, semantic slots, and linguistic targets are jointly shaped through a unified objective. The full training procedure is summarized in Algorithm 1, which outlines the data flow and optimization steps within each training epoch.

During **inference**, the *EEG Encoder* and *Semantic Decomposer* generate slot embeddings and activity scores from unseen EEG input. The *Semantic Retriever* then searches the text embedding space for the nearest semantic units, filtering by confidence thresholds to produce a retrieved set $S_{\text{retrieved}}$ and global attributes. These are passed to the *Semantic Decoder*, which constructs a structured prompt and uses a LLM to generate the final natural-language sentence. In this way, information flows seamlessly across all five components, transforming raw brain signals into coherent and semantically faithful sentences. The complete inference pipeline is detailed in Algorithm 2.

## 3 EXPERIMENTS

We evaluate BRAINMOSAIC on neural-to-semantic decoding tasks and address three questions:

**Q1:** Is communicative intent adequately modeled as a set of semantic units?

**Q2:** Does decoding in a continuous semantic space enable better generalization and scalability?

**Q3:** Does semantic-constrained generation improve fidelity and interpretability?

### 3.1 EXPERIMENT SETUP

**Datasets.** To rigorously evaluate SID and BRAINMOSAIC, we curated three public datasets based on two criteria: (1) neural signals must be segmentable into semantically complete units (i.e., sentences); (2) recordings must be collected during cognitive tasks. Their basic characteristics are summarized in Table 2, with dataset-specific details and preprocessing procedures in Appendix A. Notably, the Chisco dataset (Zhang et al., 2024) was recorded directly at the sentence level, with each trial corresponding to a complete utterance. Other public datasets contain continuous passages with character-level timestamps, which are later segmented into sentences during preprocessing. As the stimuli are presented as full sentences with self-contained semantics and focus on everyday activities, Chisco most closely aligns with our definition of Semantic Intent. In addition, we collected a private clinical SEEG dataset at a partner hospital on invasive recordings. It includes one participant performing a sentence-level imagined speech task involving 515 everyday Chinese sentences with colloquial semantics covering daily activities and interpersonal interactions. During recording, the participant completed a memory-judgment task to decide if each sentence related to a target topic. Detailed information is provided in Appendix B. Additional experiments on the clinical dataset, including brain-region contribution analyses, are provided in Appendices H and I.

**Baselines.** We adapt open-source methods and design four main baselines: (1) *Cls-Align* trains an EEG classifier using classification labels (Zhang et al., 2024). Its frozen backbone is then aligned with pre-trained sentence embeddings for cross-modal mapping. (2) *Multi-Cls* uses the same backbone as *Cls-Align* for multi-label classification, predicting top-$k$ candidate semantic units to form a complete sentence. (3) *Neuro2Semantic* aligns EEG signals to a pre-trained text embedding space, and then employs a language model to generate sentences from the aligned embeddings. It enables unconstrained text generation but is limited to English corpora (Shams et al., 2025). (4) *Seq-Decode* replaces the set-matching stage with an LSTM-based sequential decoder, while keeping the same ModernTCN encoder and LLM. This baseline is designed to assess the contribution of set matching. Implementation details are in Appendix C. Appendix C.5 further reports baselines using fully randomized labels and random/corpus-based predictions.

**Evaluation Metrics.** SID operates in a continuous, open-vocabulary space where traditional discrete metrics are inadequate. At the concept level, existing measures fail to capture whether decoded units fall into the correct semantic neighborhood. At the sentence level, we prioritize semantic fidelity rather than surface overlap; n-gram metrics (e.g., BLEU, ROUGE) emphasize exact token matches while underestimating synonymy and paraphrasing (detailed analysis in Appendix D).

To address these gaps, we design three embedding-based metrics. Unit Matching Accuracy (**UMA**) reflects hard correctness at the concept level: a predicted unit $\hat{\mathbf{z}}_i$ is counted as correct only when its similarity to the gold unit $\mathbf{z}_i^*$ exceeds a predefined threshold $\tau$. Mean Unit Similarity (**MUS**) complements this with a soft measure of alignment by averaging unit-wise similarities, capturing graded improvements even when predictions fall close to the threshold. We further define $\mathbf{MUS}_{exp}$ to quantify the semantic density within a corpus space. It is computed by uniformly and independently sampling semantic units from the corpus without frequency weighting and then averaging the embedding similarities over all sampled pairs. This value serves as a theoretical lower bound, representing the expected **MUS** inherent to the corpus itself. Sentence Reconstruction Similarity (**SRS**) evaluates sentence-level semantic fidelity by comparing the embedding of the generated sentence $\hat{\mathbf{s}}$ with that of the reference $\mathbf{s}^*$, emphasizing meaning rather than exact wording. For BRAINMOSAIC, we generate five candidate sentences using the GPT-4o-mini LLM and report the average $SRS$.

$$(1)\text{UMA} = \frac{1}{N}\sum_{i=1}^{N}\mathbb{I}\left(\text{sim}(\hat{\mathbf{z}}_i, \mathbf{z}_i^*) > \tau\right), \quad (2)\text{MUS} = \frac{1}{N}\sum_{i=1}^{N}\text{sim}(\hat{\mathbf{z}}_i, \mathbf{z}_i^*), \quad (3)\text{SRS} = \text{sim}(\hat{\mathbf{s}}, \mathbf{s}^*)$$

where $N$ denotes the total number of ground-truth semantic units, $\text{sim}(\cdot, \cdot)$ denotes a cosine-based similarity and $\mathbb{I}(\cdot)$ is an indicator function. We also report the BERTScore-F1 (Zhang* et al., 2020) as a sentence-level reference metric. In Appendix D, we provide qualitative examples for the above metrics to more intuitively illustrate the semantic relevance reflected by their values.

All experiments are conducted in-subject. For datasets with multiple subjects, we report $mean \pm std$ across subjects; for the clinical dataset, the reported variance comes from multiple runs. Unless noted otherwise, all experiments use the doubao-embedding-large (ByteDance, 2024) text space. Appendix E reports additional results with alternative and random text encoders. Regarding baseline comparisons, metric applicability varies by architecture: for **Baseline 1**, which yields only representation-level outputs without explicit sentence generation, only SRS is calculated; for **Baseline 3**, evaluation is restricted to sentence-level metrics; and for **Baselines 2 and 4**, we explicitly map their predictions into the unified text semantic space to ensure a fair comparison on MUS.

### 3.2 Q1: IS COMMUNICATIVE INTENT ADEQUATELY MODELED AS A SET OF SEMANTIC UNITS

For Question 1, we compare the set-based BRAINMOSAIC with two alternative paradigms, detailed in Table 3. Baseline *(1) Cls-Align* assigns a single, predefined topic category to each sentence, making it overly reductive for diverse expressions. For fair comparison, we first train the classification model using its original categorical framework. We then freeze its parameters and train an additional mapping module to project its outputs into the same sentence-level embedding space used by BRAINMOSAIC for similarity evaluation. *(1) Cls-Align* shows limited performance, highlighting the difficulty of capturing nuanced, compositional meaning within the constrained label sets of existing label-classification frameworks and datasets.

The second comparison examines whether modeling intent as an unordered set offers advantages over sequential decoding. Baseline *(4) Seq-Decode* replace the proposed set-matching module with

Table 3: **Comparison of baselines and BRAINMOSAIC with error bars; best results in bold. Cases with $p \leq 0.001$ are marked with *** based on a two-sided t-test against the best baseline.**

| | Clinical | | | |
|---|---|---|---|---|
| | Concept Level | | Sentence Level | |
| | **UMA** | **MUS** | **SRS** | **BERT-F1** |
| Cls-Align | – | – | $0.5976 \pm 0.0030$ | – |
| Multi-Cls | $0.0359 \pm 0.0006$ | $0.6739 \pm 0.0061$ | $0.4400 \pm$ | $0.6173 \pm$ |
| Seq-Decode | $0.1786 \pm 0.0126$ | $0.6503 \pm 0.0115$ | $0.5104 \pm$ | $0.6126 \pm$ |
| BRAINMOSAIC | $\mathbf{0.6596 \pm 0.0102}$*** | $\mathbf{0.8124 \pm 0.0108}$*** | $\mathbf{0.6651 \pm 0.0045}$*** | $\mathbf{0.6629 \pm 0.0137}$*** |

| | Chisco | | | |
|---|---|---|---|---|
| | Concept Level | | Sentence Level | |
| | **UMA** | **MUS** | **SRS** | **BERT-F1** |
| Cls-Align | – | – | $0.5040 \pm 0.0117$ | – |
| Multi-Cls | $0.0143 \pm 0.0008$ | $0.6332 \pm 0.0051$ | $0.3585 \pm 0.0063$ | $0.5692 \pm 0.0060$ |
| Seq-Decode | $0.0301 \pm 0.0016$ | $0.5503 \pm 0.0042$ | $0.5439 \pm 0.0055$ | $0.5706 \pm 0.0034$ |
| BRAINMOSAIC | $\mathbf{0.5617 \pm 0.0085}$*** | $\mathbf{0.8009 \pm 0.0024}$*** | $\mathbf{0.6206 \pm 0.0034}$*** | $\mathbf{0.6195 \pm 0.0019}$*** |

| | ChineseEEG-2 | | | |
|---|---|---|---|---|
| | Concept Level | | Sentence Level | |
| | **UMA** | **MUS** | **SRS** | **BERT-F1** |
| Cls-Align | – | – | $0.5475 \pm 0.0192$ | – |
| Multi-Cls | $0.0099 \pm 0.0002$ | $0.6058 \pm 0.0047$ | $0.3992 \pm 0.0084$ | $0.5573 \pm 0.0058$ |
| Seq-Decode | $0.0420 \pm 0.0377$ | $0.5373 \pm 0.0116$ | $0.5357 \pm 0.0073$ | $0.5586 \pm 0.0053$ |
| BRAINMOSAIC | $\mathbf{0.3707 \pm 0.0144}$*** | $\mathbf{0.7687 \pm 0.0035}$*** | $\mathbf{0.6163 \pm 0.0025}$*** | $\mathbf{0.6002 \pm 0.0031}$*** |

| | ZuCoSR | | | |
|---|---|---|---|---|
| | Concept Level | | Sentence Level | |
| | **UMA** | **MUS** | **SRS** | **BERT-F1** |
| Cls-Align | – | – | $0.4630 \pm 0.0390$ | – |
| Multi-Cls | $0.0003 \pm 0.0001$ | $0.6469 \pm 0.0038$ | $0.3385 \pm 0.0053$ | $0.4008 \pm 0.0036$ |
| Neuro2Semantic | – | – | $0.5211 \pm 0.0085$ | $0.4296 \pm 0.0230$ |
| Seq-Decode | $0.0451 \pm 0.0101$ | $0.6321 \pm 0.0061$ | $0.6171 \pm 0.0051$ | $0.4274 \pm 0.0114$ |
| BRAINMOSAIC | $\mathbf{0.7506 \pm 0.0248}$*** | $\mathbf{0.8586 \pm 0.0044}$*** | $\mathbf{0.6982 \pm 0.0026}$*** | $\mathbf{0.4640 \pm 0.0116}$*** |

| | ZuCoNR | | | |
|---|---|---|---|---|
| | Concept Level | | Sentence Level | |
| | **UMA** | **MUS** | **SRS** | **BERT-F1** |
| Multi-Cls | $0.0013 \pm 0.0012$ | $0.6393 \pm 0.0116$ | $0.2349 \pm 0.0150$ | $0.3828 \pm 0.0107$ |
| Neuro2Semantic | – | – | $0.4445 \pm 0.0057$ | $0.3865 \pm 0.0199$ |
| Seq-Decode | $0.0669 \pm 0.0141$ | $0.6104 \pm 0.0103$ | $0.5161 \pm 0.0044$ | $0.3926 \pm 0.0112$ |
| BRAINMOSAIC | $\mathbf{0.7144 \pm 0.0371}$*** | $\mathbf{0.8453 \pm 0.0043}$*** | $\mathbf{0.6094 \pm 0.0102}$*** | $\mathbf{0.4423 \pm 0.0139}$*** |

| | ZuCoTSR | | | |
|---|---|---|---|---|
| | Concept Level | | Sentence Level | |
| | **UMA** | **MUS** | **SRS** | **BERT-F1** |
| Cls-Align | – | – | $0.3797 \pm 0.0179$ | – |
| Multi-Cls | $0.0101 \pm 0.0010$ | $0.6330 \pm 0.0126$ | $0.2245 \pm 0.0099$ | $0.3753 \pm 0.0105$ |
| Neuro2Semantic | – | – | $0.4591 \pm 0.0068$ | $0.4158 \pm 0.0190$ |
| Seq-Decode | $0.0654 \pm 0.0114$ | $0.6076 \pm 0.0047$ | $0.5181 \pm 0.0079$ | $0.3860 \pm 0.0091$ |
| BRAINMOSAIC | $\mathbf{0.5520 \pm 0.0374}$*** | $\mathbf{0.8198 \pm 0.0096}$*** | $\mathbf{0.5956 \pm 0.0039}$*** | $\mathbf{0.4431 \pm 0.0100}$*** |

an LSTM decoder that predicts an ordered sequence of semantic units. Both models share the same EEG encoder and LLM module. Results show that the sequential baseline underperforms, yielding lower concept-level $UMA$ and $MUS$ and producing less coherent outputs, which underscores that overly rigid sequential modeling constrains the semantic intent expressed in EEG signals.

Table 4: BRAINMOSAIC performance under (A) vocabulary expansion and (B) training data expansion settings.

| Corpus Size | Training Ratio | Clinical | | | | | Chisco | | | | |
| --- | --- | --- | --- | --- | --- | --- | --- | --- | --- | --- | --- |
| | | UMA | MUS | $MUS_{exp}$ | SRS | BS-F1 | UMA | MUS | $MUS_{exp}$ | SRS | BS-F1 |
| *(A) Vocabulary Expansion @ Training Ratio = 1.0* | | | | | | | | | | | |
| Base | | .6596 | .8124 | *.5151* | .6651 | .6629 | .5617 | .8009 | *.4811* | .6206 | .6195 |
| Base + 500 | | .6539 | .8088 | *.5351* | .6454 | .6580 | .4603 | .8022 | *.4834* | .6184 | .6184 |
| Base + 1000 | 1.0 | .6439 | .8103 | *.5372* | .6453 | .6597 | .4595 | .8018 | *.4849* | .6152 | .6170 |
| Base + 3000 | | .6263 | .8063 | *.5399* | .6320 | .6579 | .4597 | .8019 | *.4948* | .6154 | .6143 |
| Base + 10000 | | .6276 | .8002 | *.5519* | .6295 | .6464 | .4555 | .8001 | *.5279* | .6232 | .6145 |
| Base + 30000 | | .5944 | .8042 | *.5689* | .6316 | .6393 | .4573 | .8016 | *.5660* | .6222 | .6137 |
| *(B) Training Expansion @ Word Size = Base + 30000* | | | | | | | | | | | |
| | 0.5 | .5672 | .7624 | *.5689* | .6004 | .5741 | .4337 | .8010 | *.5660* | .6220 | .6013 |
| Base + 30000 | 0.3 | .4944 | .7171 | *.5689* | .5697 | .5672 | .4292 | .7935 | *.5660* | .6083 | .5810 |
| | 0.1 | .3439 | .6963 | *.5689* | .5388 | .5479 | .4186 | .7867 | *.5660* | .5760 | .5665 |

## 3.3 Q2: DOES DECODING IN A CONTINUOUS SEMANTIC SPACE ENABLE BETTER GENERALIZATION AND SCALABILITY

**Continuity.** To answer Question 2, we first compare BRAINMOSAIC with baseline *(2) Multi-Cls*. The baseline directly classifies each input into one of the discrete semantic units from the tokenized vocabulary, which contains thousands of possible classes. For evaluation, the predicted labels are projected into the same embedding space used by BRAINMOSAIC via the shared Text Encoder, after which we apply the identical procedure to compute $UMA$ and $MUS$. The same LLM is used to reconstruct sentences from predicted units for sentence-level evaluation. Results in Table 3 show that the discrete approach performs poorly, with $UMA$ remaining extremely low and MUS close to the random expectation (see Table 6). When semantic categories grow into the hundreds or thousands, the limitations of discrete classification methods become apparent. In natural language, however, the number of semantic categories reaches the tens of thousands. In contrast, BRAINMOSAIC decodes directly into a continuous semantic space, where vector proximity encodes conceptual relatedness, enabling far greater flexibility and scalability.

**Expandability.** We conducted two experiments to evaluate the generalization and scalability of continuous semantic decoding. First, we test open-set decoding by expanding retrieval vocabulary with unseen high-frequency words, specifically the top 500, $1k$, $3k$, $10k$, $30k$ from the List of Commonly Used Words in Modern Chinese (Li & Su, 2021), on Clinical and Chisco datasets (see Table 4, A). This setting simulates realistic language use and tests whether the continuous semantic space enables effective handling of out-of-vocabulary (OOV) words. Notably, the $30k$ condition far exceeds everyday vocabulary and includes domain-specific terms that primarily appear in academic fields. Second, we examine scalability by starting with the largest vocabulary and gradually raising training data proportion from 10% up to 100% (see Table 4, B). This tests whether the semantic space is learnable and extensible without degrading performance on previously learned labels.

Results confirm that continuous semantic decoding enables strong generalization and scalability. In the open-vocabulary inference scenario (A), $UMA$ decreases only modestly as the vocabulary expands, showing robustness to a larger search space. $MUS$ and $SRS$ remain relatively stable or even improve slightly as the vocabulary expands, showing that when an exact match is missing, the model retrieves a semantically related substitute. The model's $MUS$ remains stable even as the expected random similarity ($MUS_{exp}$) increases with vocabulary density, indicating that retrieval is driven by meaningful semantic structure, not random proximity. In the training expansion scenario (B), performance scales consistently with training data's volume. While performance is limited with only 10% of the data, both $UMA$ and $SRS$ improve substantially as the training set grows. Critically, the model not only expands its active vocabulary but also learns to generalize, predicting words that never appeared in the smaller subsets. This expansion does not degrade performance on previously learned units, demonstrating that the continuous semantic space can be learned without a closed vocabulary constraint.

Table 5: Ablation Results of BRAINMOSAIC on Clinical and Chisco.

|  | Clinical | | | Chisco | | |
|---|---|---|---|---|---|---|
|  | UMA | MUS | SRS | UMA | MUS | SRS |
| w/o Set | 0.0792 | 0.7052 | 0.5721 | 0.0260 | 0.6690 | 0.5506 |
| w/o ContSpace | 0.0137 | 0.6393 | 0.4604 | 0.0044 | 0.6009 | 0.3727 |
| w/o LLM | 0.6596 | 0.8124 | 0.5456 | 0.5617 | 0.8009 | 0.5133 |
| Full Model | 0.6596 | 0.8124 | 0.6651 | 0.5617 | 0.8009 | 0.6206 |

### 3.4 Q3: DOES SEMANTIC-CONSTRAINED GENERATION IMPROVE FIDELITY AND INTERPRETABILITY

To address Question 3, we compare BRAINMOSAIC with the end-to-end generation-based baseline Neuro2Semantic to assess differences in fidelity and interpretability. Neuro2Semantic employs an LSTM-based architecture to directly translate neural signals into continuous text by mapping EEG data into a pre-trained text embedding space, enabling unconstrained sentence generation. However, a key limitation of Neuro2Semantic and similar methods is their reliance on the language of the pre-trained generator. As the released model is English-only, our comparison is confined to the English ZuCo datasets. BRAINMOSAIC consistently outperforms the baseline, confirming that constraining generation with decoded semantic units significantly improves fidelity to the intended meaning. Furthermore, the semantic-scaffolded strategy in BRAINMOSAIC enhances interpretability by providing transparent intermediate semantic units. Reliability tests using various LLMs and prompt variations confirmed consistent performance advantages, validating semantic-constrained generation robustly improves fidelity and maintains stable interpretable decoding. Details are in Appendix F.

### 3.5 ABLATION ANALYSIS OF BRAINMOSAIC

To assess the contribution of each component in BRAINMOSAIC, we conduct a series of ablation experiments. **(a) Ablation of the Semantic Decomposer (w/o Set)**. The EEG encoder is directly aligned with sentence embeddings without the set-based decomposition stage. During inference, the model selects the top-$K$ semantic units most similar to the predicted sentence embedding, where $K$ equals to the number of slots in BRAINMOSAIC. **(b) Ablation of the Continuous Semantic Space (w/o ContSpace).** The semantic retrieval module is replaced by a multi-label classifier. The model directly predicts $K$ category labels, which serve as the semantic units without leveraging the continuous embedding space. **(c) Ablation of the LLM Semantic Decoder (w/o LLM).** Given the semantic unit set produced by BRAINMOSAIC, no language model is used for sentence reconstruction. Instead, the units are concatenated in random orders to form 50 candidate sentences per instance, and the reported metrics use the highest SRS for each instance.

As shown in Table 5, performance decreases in all three ablation settings, demonstrating that each component is essential to BRAINMOSAIC. Removing the set-based semantic decomposer forces the model to align EEG signals only with sentence-level embeddings. However, sentence-level semantics are highly entangled, and their relationships lack the structured organization as semantic units. Ablating the continuous semantic space yields behavior similar to baseline *(2) Multi-Cls*, as the model degenerates into a multi-label classifier. In the LLM ablation, the predicted semantic units remain unchanged and only the sentence reconstruction stage is removed. The results indicate that the decoded semantic units are informative but fragmented. The LLM performs this post-processing automatically, organizing the units into fluent and interpretable outputs. This highlights the LLM's role in transforming conceptual predictions into natural language.

## 4 DISCUSSION

We introduced Semantic Intent Decoding (SID) and its implementation BRAINMOSAIC, a transparent compositional framework for translating neural signals into natural language. Moving beyond fixed-label classification and opaque end-to-end models, it enables more natural BCI communication and advances study of neural meaning representations. Future work can improve semantic decomposition, explore intent representation across brain regions, and extend SID to new modalities and real-time systems. These efforts will advance BCIs to more accurate flexible systems, fostering new AI-neuroscience intersection discoveries.

## ACKNOWLEDGEMENT

This work is supported by NSFC (62322606, 62441605).

## ETHICS STATEMENT

All experiments were conducted in strict compliance with relevant ethical guidelines and institutional review processes. The private clinical SEEG dataset was collected under approved IRB protocols at a partner hospital, with informed consent obtained from the participant. No personally identifiable information is included. To protect privacy and security, the private dataset will not be publicly released, and only aggregate statistics and example stimuli are provided in the Appendix B.

## REPRODUCIBILITY STATEMENT

To ensure reproducibility, comprehensive descriptions of model architectures, training configurations, and evaluation protocols are provided in the main text and appendix. Our source code and experiment scripts are included in GitHub repository (`https://github.com/Erikaqvq/BrainMosaic_ICLR26`) to facilitate replication. Due to privacy restrictions, the private clinical dataset cannot be fully released; however, detailed documentation and representative examples are included in the Appendix B to aid understanding and replication with comparable data.

## USE OF LARGE LANGUAGE MODELS

Large Language Models were used solely to aid in the polishing of language and writing clarity during manuscript preparation.

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

# A   PUBLIC DATASET

Table 6: Summary of datasets used in our experiments. *Sent.* represents Sentence. *OOV* denotes the proportion of concepts that appear in the test set but not in the training set.

| Dataset | Task (#Subjects) | #Sents | #Words | OOV | $MUS_{exp}$ (Doubao) | $MUS_{exp}$ (Qwen) | Signal | #Channels |
|---|---|---|---|---|---|---|---|---|
| Chisco | Read (5), Imagine (5) | 6,681 | 4,595 | .3026 | .4855 | .4050 | EEG | 122 |
| ChineseEEG2 | Read (4), Listen (8) | 3,621 | 3,910 | .3663 | .4715 | .4144 | EEG | 128 |
| ZuCo 1.0 | Read (12) | 1,039 | 5,398 | .6668 | .6062 | .4354 | EEG | 105 |
| ZuCo 2.0 | Read (18) | 663 | 3,461 | .6431 | .6083 | .4316 | EEG | 105 |
| Private | Imagine (1) | 515 | 350 | .1215 | .5244 | .4567 | SEEG | 132 |

Table 6 provides an overview of the datasets, covering both Chinese and English and spanning tasks from silent reading to imagined speech, as well as different signal modalities. Following standard practice in neural decoding, all experiments were conducted under an in-subject setting, where a separate model is trained and evaluated for each individual subject. For each subject, all trials were randomly split into an 8:2 train-test ratio and repeated multiple times. For subjects who participated in multiple experimental paradigms within the same project (e.g., both listening and reading tasks), we treat each paradigm as an independent subject. Across all datasets, each subject completed the experiment only once. To further analyze the linguistic characteristics of these corpora, we computed the OOV ratio, defined as the proportion of words appearing in the test set but absent in the training set relative to the testing vocabulary size. This measure reflects the lexical generalization challenge across datasets. In addition, we incorporated two widely used two embedding models, *Doubao-embedding-large* and *Qwen3-Embedding-8B*, and calculated the expected inter-words similarity. These statistics quantify corpus-level redundancy and semantic diversity from the perspective of modern sentence embeddings.

## A.1   CHISCO

Chisco (Zhang et al., 2024) is a large-scale EEG corpus for imagined speech decoding, containing recordings from 5 subjects with over 20,000 samples, and about 900 minutes per subject. The stimuli cover over 6,000 phrases across 39 semantic categories. Each trial consists of a Reading Phase (sentence memorization) and a Recall Phase (imagined speech). We used the preprocessed data, where the continuous recordings were further segmented into a 5s reading segment and a 3.3s imagined speech segment based on event markers.

## A.2   CHINESEEEG

ChineseEEG-2o (Chen et al., 2025) is a naturalistic EEG dataset collected while participants either read aloud or passive listen to two full-length novels, the Little Prince and Garnett Dream, both in Chinese version. In general, the text materials consisted of 46,591 Chinese characters. To ensure linguistic coherence, the data acquisition preserved the line-level integrity of the texts. Following the dataset protocol, we segment the continuous EEG recordings according to the line-level text labels, aligning each trial with the corresponding sentence fragment.

## A.3   ZUCO

ZuCo is a multimodal corpus combining high-density EEG and eye-tracking during natural reading. It includes two versions. ZuCo 1.0 (Hollenstein et al., 2018), with data from 12 English native speakers reading about 4 - 6 hours of text, and ZuCo 2.0 (Hollenstein et al., 2019), which expands the corpus with 739 sentences from 18 participants. The experimental design covers three tasks: (1) normal reading with sentiment analysis on movie reviews (SR), (2) normal reading with comprehension questions on Wikipedia sentences (NR), and (3) task specific reading (TSR) requiring relation type identification (e.g., award, education). In our work, we used the preprocessed EEG data, aligned with the corresponding text stimuli based on event markers.

## B  THE CLINICAL DATASET

The intracranial EEG dataset was recorded using 12 electrode arrays comprising 132 channels, providing broad coverage across cortical and subcortical regions. This comprehensive electrophysiological mapping was designed to advance our understanding of neural mechanisms underlying conceptual judgment during perceptual processing.

**Participants.**  A male individual diagnosed with epilepsy participated in our study. The participant underwent implantation of iEEG electrodes as part of his clinical treatment for epilepsy. The participant voluntarily enrolled in the study and provided written informed consent. The experimental protocol and data collection procedures were reviewed and approved by the institutional review board of Huashan Hospital, Fudan University.

**Tasks.**  Five conceptual categories were selected, with 200 corresponding sentences assigned to each. Examples of sentences across these categories are provided in Table 7. During the experiment, participants judged whether the presented concept matched the sentence and indicated their response via keypress.

**Electrode Localization and Visualization.**  The number and positioning of the electrodes were selected based on clinical requirements. Anatomical designations were obtained through Freesurfer's cortical parcellation.

**Neural signal recording.**  Neural activity was recorded using a multi-channel electrophysiological recording system (EEG-1200C, Nihon Kohden, Tokyo, Japan) with a sampling rate of 2000 Hz.

**Data preprocessing.**  The neural signals were extracted based on the data labels recorded during the experiment, and then downsampled from 2000Hz to 1000 Hz and notch filtered at 50 Hz, 100 Hz, and 150 Hz. All neural signals underwent bipolar referencing and normalization.

Table 7: Representative Chinese sentences and their English translations in the Clinical dataset.

| Category | Chinese Sentence | English Translation |
|---|---|---|
| Dining | 饭菜好香啊！
肚子咕咕叫，吃点东西吧。 | The food smells so good!
My stomach is growling, let's eat something. |
| Phone | 手机里的东西真有趣。
我想玩手机，放松一下。 | The things on the phone are really interesting.
I want to play with my phone to relax. |
| Sleeping | 我困了，想睡觉了。
早点睡，明天有精神。 | I'm sleepy and want to go to bed.
Go to bed early, and you'll be energetic tomorrow. |
| Toilet | 厕所堵了，怎么办？
我肚子不舒服，得上厕所。 | The toilet is clogged, what should I do?
My stomach feels uncomfortable, I need to go to the toilet. |
| Health Care | 请帮我拿药。
我对这药有过敏反应。 | Please help me get the medicine.
I have an allergic reaction to this medicine. |

# C  BASELINE

## C.1  CLS-ALIGN

This baseline follows the approach of Chisco, employing EEGNet (Lawhern et al., 2018) as the backbone to extract low-level EEG features, which are subsequently processed by a transformer encoder with self-attention and a fully connected layer to capture higher-level representations. The training process consists of two stages. First, a classification task is performed, in which neural signals are categorized based on dataset-specific labels. These labels include 39 classes in Chisco, 2 classes for book source in ChineseEEG-2, 3 classes for sentiment in ZuCo-SR, 7 - 9 classes for relation type identification in ZuCo-TSR, and 5 classes for Clinical. It is worth noting that for ZuCo-NR dataset, patients are required to complete multiple-choice questions rather than producing categorical labels, making such classification-based training infeasible. After the classification training, the backbone is frozen, and a projection head is trained to align neural embeddings with corresponding sentence embeddings. The final similarity is evaluated between the generated neural embeddings and the textual embeddings.

## C.2  MULTI-CLS

This baseline adopts the same backbone as Cls-Align (EEGNet + transformer encoder) to extract neural representations. Instead of aligning with text embeddings, the model directly performs multi-class classification, where the predicted top-k class labels are treated as concept-level cluster tags. These predicted tags are then used to query a large language model (LLM), allowing the system to generate natural language sentences from neural signals.

## C.3  NEURO2SEMANTIC

This model uses an LSTM adapter as the backbone to encode neural signals into fixed-dimensional embeddings aligned with pre-trained text embeddings (text-embedding-ada-002) (Shams et al., 2025). Training proceeds in two phases: (1) in the alignment phase, the LSTM adapter is trained with a combination of contrastive loss and triplet margin loss to enforce semantic consistency between neural signals and text embeddings; (2) in the decoding phase, the adapter is frozen, and a Vec2Text inversion module (a transformer-based corrector) is fine-tuned to reconstruct coherent text sequences from the aligned neural embeddings. Through this two-step process, the model is able to directly generate natural language sentences from neural signals.

## C.4  SEQ-DECODE

This baseline employs ModernTCN (Luo & Wang, 2024) as the neural signal encoder, which modernizes 1D convolution blocks by decoupling temporal, feature, and variable dimensions and stacking them in a residual architecture to capture both cross-time and cross-variable dependencies. On top of this encoder, an LSTM-based sequential decoder predicts a sequence of concept labels step by step, with an internal binary classifier to determine whether to stop prediction. Finally, the predicted concept label sequence is passed to an LLM, which generates coherent sentences by filling in the semantic content in the predicted order.

## C.5  RANDOM BASELINES

To more clearly demonstrate the contribution of the EEG semantic decoding module, we compare the proposed BRAINMOSAIC against label-level and prediction-level random baselines under the Doubao embedding setting.

**Label-level Random baseline** preserves the EEG semantic-unit decoding module. For each sample, we randomly sample $k$ semantic units from the corpus, where $k$ equals the number of true units in that sample's golden sentence, and compare the decoded predictions with these randomly sampled pseudo-labels using UMA and MUS. For the sentence-level metric, SRS is computed between the model-generated sentence and a randomly selected sentence from the corpus. This baseline evaluates whether the model's predictions truly reflect the semantic intent contained in the input EEG, rather than matching by chance.

Table 8: Random and text-prior baselines compared with BRAINMOSAIC across datasets.

| | Clinical | | | Chisco | | | ChineseEEG-2 | | |
|---|---|---|---|---|---|---|---|---|---|
| | UMA | MUS | SRS | UMA | MUS | SRS | UMA | MUS | SRS |
| Label Random | 0.2472 | 0.6878 | 0.5394 | 0.1390 | 0.6987 | 0.4997 | 0.1308 | 0.6955 | 0.4683 |
| Random-prior | 0.0563 | 0.6392 | 0.5056 | 0.0152 | 0.6185 | 0.4924 | 0.0112 | 0.5815 | 0.5129 |
| Topk-prior | 0.1510 | 0.6884 | 0.5418 | 0.1215 | 0.6793 | 0.5465 | 0.1159 | 0.6606 | 0.5282 |
| Freq-prior | 0.1008 | 0.6540 | 0.5321 | 0.0289 | 0.6365 | 0.5124 | 0.0332 | 0.6046 | 0.5170 |
| BRAINMOSAIC | **0.6596** | **0.8124** | **0.6651** | **0.5617** | **0.8009** | **0.6206** | **0.3707** | **0.7687** | **0.6163** |

| | ZuCoSR | | | ZuCoNR | | | ZuCoTSR | | |
|---|---|---|---|---|---|---|---|---|---|
| | UMA | MUS | SRS | UMA | MUS | SRS | UMA | MUS | SRS |
| Label Random | 0.2557 | 0.7591 | 0.5499 | 0.2640 | 0.7339 | 0.5297 | 0.2588 | 0.7084 | 0.5009 |
| Random-prior | 0.0624 | 0.7198 | 0.5526 | 0.0788 | 0.7163 | 0.4879 | 0.0641 | 0.7049 | 0.4999 |
| Topk-prior | 0.1559 | 0.7407 | 0.5681 | 0.1467 | 0.7554 | 0.5227 | 0.1266 | 0.7137 | 0.5033 |
| Freq-prior | 0.1078 | 0.7311 | 0.5612 | 0.1003 | 0.7351 | 0.5033 | 0.0951 | 0.7190 | 0.4916 |
| BRAINMOSAIC | **0.7506** | **0.8586** | **0.6982** | **0.7144** | **0.8453** | **0.6094** | **0.5520** | **0.8198** | **0.5956** |

**Prediction-level Random baselines** bypass the EEG decoding module and directly sample the prediction set from the corpus. For each sample, we select $k$ semantic units (corresponding to the number of model slots), compare the resulting UMA and MUS with the golden labels, and then use an LLM to synthesize sentences from this randomly sampled set to compute SRS against the golden sentence. We provide three text-prior strategies: (a) **Random-prior**: randomly select $k$ semantic units from the corpus; (b) **TopK-prior**: deterministically take the top-$k$ most frequent units in the corpus; (c) **Freq-prior**: randomly select $k$ semantic units according to frequency distribution. These baselines estimate the performance that can be achieved solely from corpus statistics or language priors, without using EEG information at all.

The results in Table 8 show that all random and text-prior baselines perform poorly within the SID framework. Although baselines that exploit corpus priors may occasionally obtain deceptively high concept-level scores, the resulting semantic sets are inherently meaningless and collapse to chance level when evaluated at sentence level. These findings further indicate that the model's outputs genuinely reflect the semantic information decoded from EEG, rather than artifacts of corpus statistics or LLM priors.

# D ANALYSIS AND QUALITATIVE EXAMPLES OF EVALUATION METRICS

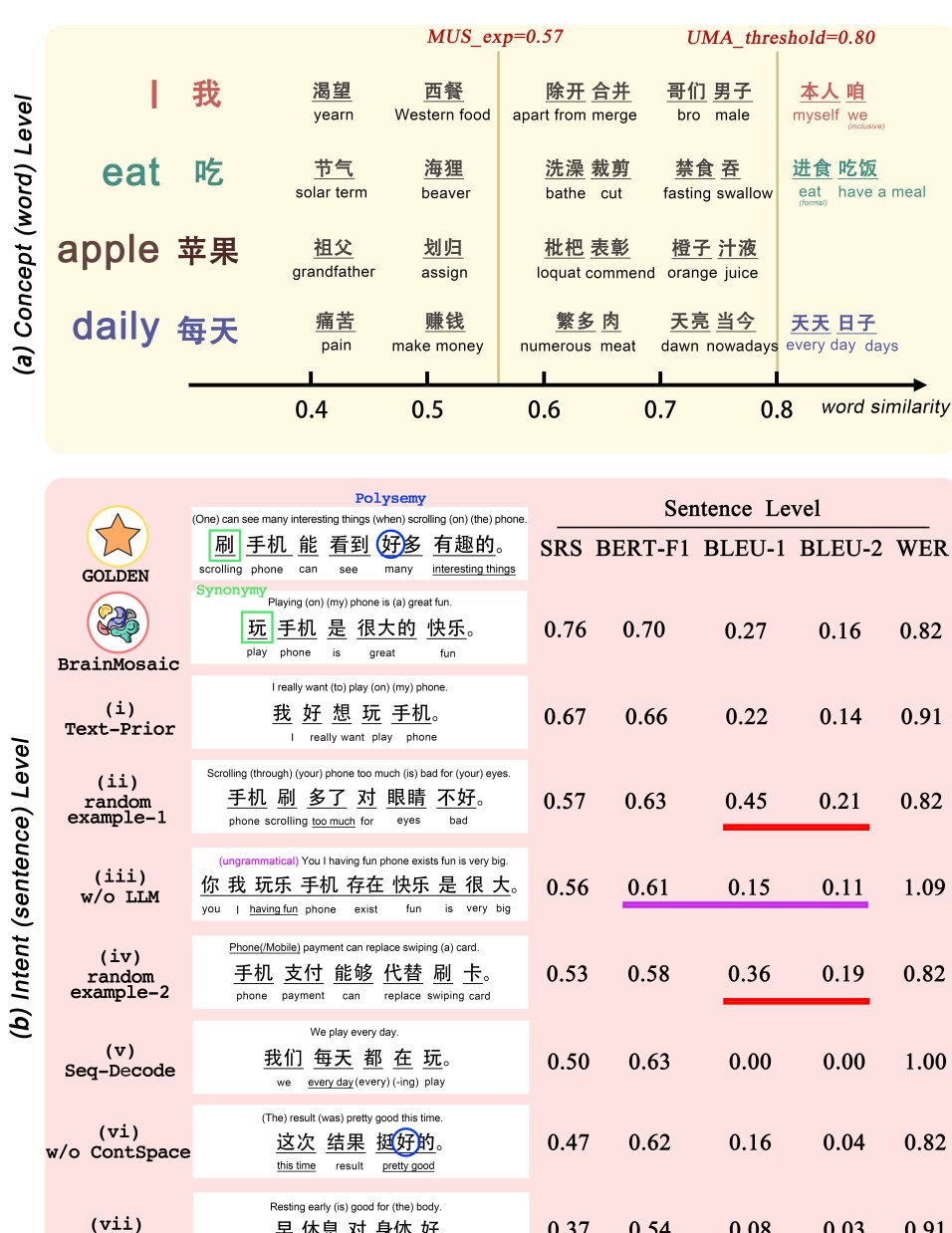

Figure 2: **Qualitative examples of two-level evaluation metrics.** Underlines mark meaningful Chinese or English word groups. (Parentheses indicate additional elements inserted in English translations for syntactic or semantic clarity.) **(a) Concept (word) level examples** are sampled from the union of the Clinical + Chisco + the 30k Chinese dictionary. Each word falls into one of the embedding similarity intervals [0, 0.5), [0.5, 0.6), [0.6, 0.7), [0.7, 0.8), [0.8, 1.0) relative to the target units in the Doubao embedding space. **(b) Intent (sentence) level examples** including the gold reference, one prediction from BRAINMOSAIC, five baseline results and two manually constructed sentences, illustrating typical linguistic phenomena (sorted by SRS score in descending order).

## D.1 Evaluation Metric Design

To comprehensively evaluate the SID framework, we assess model outputs at two levels: the concept (word) level and the intent (sentence) level. This ensures an objective and accurate assessment of both the decoding accuracy of individual semantic units and the extent to which the reconstructed sentences restore the actual semantic intent behind the brain signals. Traditional Natural Language Processing (NLP) token-level metrics aim for strict, precise comparison of surface characters, which are more suitable for tasks like Speech Decoding (e.g., decoding Chinese initials/finals or English syllables). In contrast, the SID framework explores human cognition, where the understanding and expression of semantics are inherently abstract. Consequently, SID and the broader concept decoding paradigm require metrics that evaluate relative semantic relationships rather than character-level exact matches. For this reason, we introduced the Unit Matching Accuracy (UMA), Mean Unit Similarity (MUS), and Sentence Reconstruction Similarity (SRS) to capture semantic intent beyond surface forms.

## D.2 Qualitative Analysis of Concept-Level Metrics

Figure 2(a) illustrates the applicability of concept-level metrics using the sentence "I eat apples everyday" as a reference. We analyzed the semantic similarity between the four semantic units in this sentence and all words from a large Chinese corpus. The results demonstrate that when UMS exceeds 0.8, the semantic correlation between words is significant. For instance, the word "eat" (吃) matches strongly with "eat" (进食) and "have a meal" (吃饭), which represent the same action in formal and casual contexts, respectively. Conversely, "apple" (苹果) lacks any synonym with a similarity greater than 0.8, reflecting that "apple" is a specific entity without close synonyms in this Chinese corpus. Based on these observations, we set the cluster threshold and UMA threshold at 0.8 in our experiments. This effectively clusters semantically identical or highly related terms (e.g., connecting "myself" (本人) and "we" (咱) with "I" (我)) while avoiding an overly bloated semantic space. Furthermore, the MUS_exp (the expected embedding similarity of two randomly drawn semantic units) is approximately 0.57. As shown in the interval [0.5, 0.6], word pairs near this value share almost no meaningful semantic or contextual connection, confirming that our metrics effectively distinguish between random noise and true semantic relatedness.

## D.3 Qualitative Analysis of Intent-Level Metrics

Figure 2(b) provides a qualitative comparison of sentence-level metrics, highlighting the limitations of traditional NLP metrics (like BLEU and WER) for SID tasks. We compared the ground truth (Golden) sentence "One can see many interesting things when scrolling on the phone" (刷手机能看到好多有趣的) with the prediction from BRAINMOSAIC and several baselines. The BRAIN-MOSAIC output, "Playing on my phone is a great fun" (玩手机是很大的快乐), captures the core intent - using a phone and experiencing enjoyment - despite surface-level differences. Traditional metrics fail to capture this semantic alignment due to four primary linguistic phenomena common in languages:

**1. Polysemy (One Word, Multiple Meanings):** Standard metrics cannot distinguish context-dependent meanings. For example, the character "好" in the Golden sentence appears in "好多" meaning "many." However, in sentence (vi), "The result was pretty good this time" (这次结果挺好的), and sentence (vii), "Resting early is good for the body" (早休息对身体好), the same character "好" means "good." Similarly, "刷" in the Golden sentence means "scrolling," whereas in sentence (iv), "Phone payment can replace swiping a card" (手机支付能够代替刷卡), it means "swiping" a card. BLEU-based metrics reward these character matches despite the different meanings.

**2. Synonymy (One Meaning, Multiple Words):** BRAINMOSAIC uses "玩" (play) instead of "刷" (scroll) to describe using a phone. While "刷" is more vivid, both verbs functionally describe the same activity in this context. Additionally, the model uses "很大" (very big/great) to express high degree, paralleling "好多" (many) in the Golden sentence. Since Chinese does not strictly distinguish countability for degree adverbs, these are semantically equivalent, yet rigid token matching penalizes this variation.

Table 9: Supplementary Metrics across datasets.

| | Clinical | | | | Chisco | | | |
|---|---|---|---|---|---|---|---|---|
| | **BLEU1** | **BLEU2** | **BLEU4** | **WER** | **BLEU1** | **BLEU2** | **BLEU4** | **WER** |
| Multi-Cls | 0.1291 | 0.0550 | 0.0257 | **0.9344** | 0.0530 | 0.0195 | 0.0112 | **0.9949** |
| Seq-Decode | 0.1231 | 0.0464 | 0.0231 | 0.9445 | 0.0453 | 0.0159 | 0.0091 | 1.0438 |
| BRAINMOSAIC | **0.1787** | **0.0841** | **0.0393** | 0.9635 | **0.0947** | **0.0391** | **0.0190** | 1.0361 |

| | ChineseEEG-2 | | | | ZuCoSR | | | |
|---|---|---|---|---|---|---|---|---|
| | **BLEU1** | **BLEU2** | **BLEU4** | **WER** | **BLEU1** | **BLEU2** | **BLEU4** | **WER** |
| Multi-Cls | 0.0420 | 0.0156 | 0.0089 | 1.0896 | 0.0046 | 0.0016 | 0.0013 | **0.8900** |
| Neuro2Semantic | – | – | – | – | 0.0689 | **0.0175** | **0.0081** | 1.2158 |
| Seq-Decode | 0.0496 | 0.0197 | 0.0106 | 1.1675 | 0.0049 | 0.0018 | 0.0014 | **0.8543** |
| BRAINMOSAIC | **0.0813** | **0.0299** | **0.0158** | **1.0747** | **0.0754** | 0.0163 | **0.0081** | 1.0618 |

| | ZuCoNR | | | | ZuCoTSR | | | |
|---|---|---|---|---|---|---|---|---|
| | **BLEU1** | **BLEU2** | **BLEU4** | **WER** | **BLEU1** | **BLEU2** | **BLEU4** | **WER** |
| Multi-Cls | 0.0045 | 0.0017 | 0.0013 | 0.9197 | 0.0043 | 0.0017 | 0.0013 | 0.8954 |
| Neuro2Semantic | **0.0721** | **0.0199** | **0.0086** | 1.0940 | **0.0901** | **0.0247** | **0.0096** | 1.0971 |
| Seq-Decode | 0.0114 | 0.0043 | 0.0031 | **0.8646** | 0.0083 | 0.0035 | 0.0023 | **0.8842** |
| BRAINMOSAIC | 0.0556 | 0.0167 | 0.0082 | 1.0594 | 0.0543 | 0.0170 | 0.0081 | 1.0520 |

**3. Keyword Proximity without Semantic Relevance:** Sentence (ii), (iv) contain high keyword overlap ("手机", "刷") with the Golden sentence. However, their intents are unrelated to the Golden intent. Traditional metrics often overscore these sentences due to n-gram overlap.

**4. Broken Logic and Syntax:** Sentence (iii) represents the result of ablation study without LLM post-processing: "You I having fun phone exists fun is very big" (你我玩乐手机存在快乐是很大). This output is a concatenation of keywords without grammatical structure. While it hits several key terms, it lacks coherent logic. Our SRS metric correctly penalizes such scenario, whereas n-gram precision metrics might assign it a high score.

## D.4 METRIC SELECTION

While we observed that BERT-F1 generally aligns with SID evaluation requirements, it lacks sufficient discrimination ability. For instance, sentence (vi), which contains abstract and vague references ("this time", ""esult"), achieves a BERT-F1 score of 0.62, which is deceptively high for a sentence with low semantic relevance to the specific Golden intent. In contrast, our SRS metric better reflecting the semantic divergence. Therefore, we adopt SRS as our primary sentence-level metric, while providing BLEU and WER scores for reference in Table 9.

Table 10: Performance of different text encoder configurations on Clinical and Chisco datasets.

| | Clinical | | | | Chisco | | | |
|---|---|---|---|---|---|---|---|---|
| | **UMA** | **MUS** | **SRS** | **BERT-F1** | **UMA** | **MUS** | **SRS** | **BERT-F1** |
| Random | 0.0644 | 0.6962 | 0.5067 | 0.5962 | 0.0417 | 0.6882 | 0.4932 | 0.5719 |
| Qwen3 (Word) | 0.1994 | 0.7194 | 0.6014 | 0.6166 | 0.1601 | 0.7073 | 0.6044 | 0.5873 |
| Qwen3 (Definition) | 0.2082 | 0.7389 | 0.6271 | 0.6249 | 0.1685 | 0.7108 | 0.6121 | 0.5904 |
| Doubao (Word) | 0.6373 | 0.8014 | 0.6578 | 0.6480 | 0.5502 | 0.7986 | 0.6098 | 0.6022 |
| Doubao (Definition) | **0.6596** | **0.8124** | **0.6651** | **0.6629** | **0.5617** | **0.8009** | **0.6206** | **0.6195** |

# E ANALYSIS OF TEXT ENCODER VARIANTS

The construction of a continuous semantic space is fundamental to the performance of the SID framework. To systematically evaluate this impact, we conducted ablation studies comparing different embedding models, construction strategies, and a randomized baseline. For the Random Embedding condition, we generated a fixed 256-dimensional embedding library where vector similarities were distributed within the $(0, 1)$ interval, with a mean of approximately $0.5$ and a standard deviation of $0.1$. Each semantic unit and sentence in the corpus was assigned a fixed random vector from this library. This setup served as a control to measure model performance in the complete absence of semantic structure. We evaluated two multi-language embedding models, Qwen (Zhang et al., 2025b) and Doubao (ByteDance, 2024). We used two embedding generation strategies to assess the quality of the resulting semantic spaces:

**1. Word-based Generation:** The embedding is generated directly from the semantic unit itself (e.g., inputting "orchard" $\rightarrow get_{embedding}$("orchard")).

**2. Definition-based Generation:** The embedding is generated from the dictionary definition or semantic explanation of the unit (e.g., inputting "orchard" $\rightarrow get_{embedding}$("orchard: an intentional plantation of trees or shrubs that is maintained for food production")).

The experimental results (see Table 10) demonstrate that a well-structured text semantic space is critical for SID. The Random Embedding condition resulted in a catastrophic degradation of performance, confirming that the random initialization completely destroys the semantic structure between concepts. This validates that our model's performance relies on the intrinsic geometric relationships within the semantic space rather than fitting arbitrary high-dimensional vectors. Comparisons between the generation strategies reveal that Definition-based Generation significantly outperforms the direct Word-based approach. Current LLMs are optimized for processing and understanding longer contexts. Therefore, providing a descriptive definition allows the embedding model to capture richer semantic features than a single token can provide. Furthermore, this implies a promising scaling property for our framework: as LLM embedding technologies continue to advance in expressiveness and granularity, our method will naturally inherit these improvements, potentially leading to further gains in decoding accuracy without architectural changes.

Furthermore, we compare the performance of BRAINMOSAIC against baselines under both the Doubao (definition-based) and Qwen3 (definition-based) embedding settings. Since the choice of embedding space primarily affects the MUS metric while other metrics remain largely unchanged, we report only the two baselines that support MUS evaluation. The results in Table 11 show that, regardless of the embedding model used, BRAINMOSAIC consistently and substantially outperforms all baselines.

Table 11: Comparison of MUS results under Doubao and Qwen3 embedding spaces.

| | Clinical | | Chisco | | ChineseEEG-2 | |
|---|---|---|---|---|---|---|
| | MUS-D | MUS-Q | MUS-D | MUS-Q | MUS-D | MUS-Q |
| Multi-Cls | 0.6739 | 0.5148 | 0.6332 | 0.4508 | 0.6058 | 0.4818 |
| Seq-Decode | 0.6503 | 0.6149 | 0.5503 | 0.4728 | 0.5373 | 0.5077 |
| BRAINMOSAIC | **0.8124** | **0.7389** | **0.8009** | **0.7108** | **0.7687** | **0.7025** |

| | ZuCoSR | | ZuCoNR | | ZuCoTSR | |
|---|---|---|---|---|---|---|
| | MUS-D | MUS-Q | MUS-D | MUS-Q | MUS-D | MUS-Q |
| Multi-Cls | 0.6469 | 0.3796 | 0.6393 | 0.3571 | 0.6330 | 0.3505 |
| Seq-Decode | 0.6321 | 0.4948 | 0.6104 | 0.4924 | 0.6076 | 0.4872 |
| BRAINMOSAIC | **0.8586** | **0.7052** | **0.8453** | **0.6868** | **0.8198** | **0.6897** |

Table 12: Effect of LLM, number of input tokens ($k$), and prompt design on $SRS$. Here, $K \approx 4$ is the average number of tokens per sentence in the Clinical dataset.

| LLM | Prompt | $k = K$ | $k = 2K$ | $k = 5K$ | $k = all$ |
|---|---|---|---|---|---|
| GPT-4o-mini | with-score | 0.6876 | 0.6696 | 0.6413 | 0.6288 |
| | no-score | 0.6836 | 0.6685 | 0.6464 | 0.6305 |
| Doubao-Seed | with-score | 0.6840 | 0.6616 | 0.6329 | 0.6197 |
| | no-score | 0.6954 | 0.6823 | 0.6436 | 0.6266 |
| DeepSeek | with-score | 0.6988 | 0.6810 | 0.6444 | 0.6286 |
| | no-score | 0.6873 | 0.6726 | 0.6430 | 0.6324 |

# F    ANALYSIS OF SEMANTIC DECODER VARIATIONS

We further evaluated the impact of prompt design and parameter choices on sentence reconstruction using our clinical dataset, in which each sentence contains an average of $K = 4.365$ tokens. Our analysis considered three factors: the language model backend, the number of predicted words ($k$) provided to the model, and whether the predicted probability scores were explicitly included in the prompt. We compared three representative LLMs, GPT-4o-mini, Doubao-Seed-1.6-Flash, and DeepSeek-Chat, and tested different values of $k$ : $K, 2K, 5K$, and all predicted words with confidence scores above $0.8$.

As shown in Table 12, $SRS$ remained consistently high across all conditions, indicating that each LLM could faithfully reconstruct sentences by preserving key predicted tokens and generating coherent fillers. However, performance declined slightly when too many tokens were provided, suggesting that redundant or noisy inputs reduce fidelity. Including probability scores in the prompt produced only marginal improvements, implying that LLMs already infer token importance internally. Overall, concise and accurate token sets proved more important than prompt complexity or the choice of backend model, with all three LLMs exhibiting strong and comparable reconstruction performance.

Table 13: Effect of the semantic slot number $K$ on performance. $\bar{K}$ denotes the average number of semantic units per sentence in each dataset.

| $K$ | Clinical ($\bar{K} = 4.365$) | | | | Chisco ($\bar{K} = 4.656$) | | | |
|---|---|---|---|---|---|---|---|---|
| | **NUS** | **UMA** | **MUS** | **SRS** | **NUS** | **UMA** | **MUS** | **SRS** |
| 1 | 0.00 | 0.1493 | 0.8023 | 0.5123 | 0.00 | 0.0907 | 0.7503 | 0.4988 |
| 2 | 0.00 | 0.3433 | 0.8208 | 0.5891 | 0.00 | 0.2706 | 0.7921 | 0.5772 |
| 5 | 0.61 | 0.6290 | 0.8063 | 0.6499 | 1.59 | 0.5299 | 0.7977 | 0.5937 |
| 10 (default) | 5.36 | 0.6596 | 0.8124 | 0.6651 | 5.75 | 0.5617 | 0.8009 | 0.6206 |
| 15 | 10.35 | 0.6559 | 0.8162 | 0.6624 | 11.38 | 0.5590 | 0.7989 | 0.6172 |
| 25 | 20.64 | 0.6418 | 0.8104 | 0.6683 | 21.23 | 0.5577 | 0.8012 | 0.6208 |
| 50 | 45.56 | 0.6482 | 0.8138 | 0.6636 | 45.76 | 0.5601 | 0.7954 | 0.6160 |

# G    ANALYSIS OF THE SEMANTIC SLOT NUMBER K

To examine how the number of semantic slots $K$ influences the model performance, we conduct sensitivity analysis on the Clinical and Chisco datasets. The average number of semantic units per sentence, denoted as $\bar{K}$, is $4.365$ and $4.656$ respectively. We evaluate slot numbers $K \in \{1, 2, 5, 10 \text{ (default)}, 15, 25, 50\}$, and report the three main metrics along with the number of unused slots (NUS) during testing.

As shown in Table 13, when $K$ is smaller than the actual semantic content $\bar{K}$, the model's performance is clearly constrained by the insufficient number of slots. In contrast, when $K$ exceeds the required semantic capacity, the performance remains stable, demonstrating that the robustness of the model to the choice of slot number. This stability suggests that the designed "no-object" mechanism in the semantic retriever can effectively determine how many semantic units are actually needed.

Table 14: Five-way topic classification on the clinical dataset.

| Method | Accuracy |
|---|---|
| BRAINMOSAIC + LLM selection | **61%** |
| 5-class classifier | 44% |
| Chance level | 27% |

# H    TOPIC CLASSIFICATION ON THE CLINICAL DATASET

During topic-guided data collection, our private Clinical corpus was annotated into five semantic categories: *Healthcare*, *Sleep*, *Phone Use*, *Toilet*, and *Diet*. To compare classification strategies, we conducted two experiments: (i) using BRAINMOSAIC to generate five candidate sentences for each sample and prompting an LLM to select the most likely topic (five-way selection), and (ii) directly training a five-class classifier using the same ModernTCN backbone. As shown in Table 14, the LLM selection based on BRAINMOSAIC's outputs achieved an accuracy of **61%**, significantly higher than the **44%** obtained by the direct classifier. Since the five topics are not evenly distributed, the chance level is approximately **27%**. These results indicate that BRAINMOSAIC's intermediate semantic representations capture topic-relevant meaning more effectively than direct label supervision. This further indicates that decomposing a sentence into a set of semantic units enhances the model's ability to capture and express the underlying complex meaning.

# I    REGIONAL CONTRIBUTION ANALYSIS ON THE CLINICAL DATASET

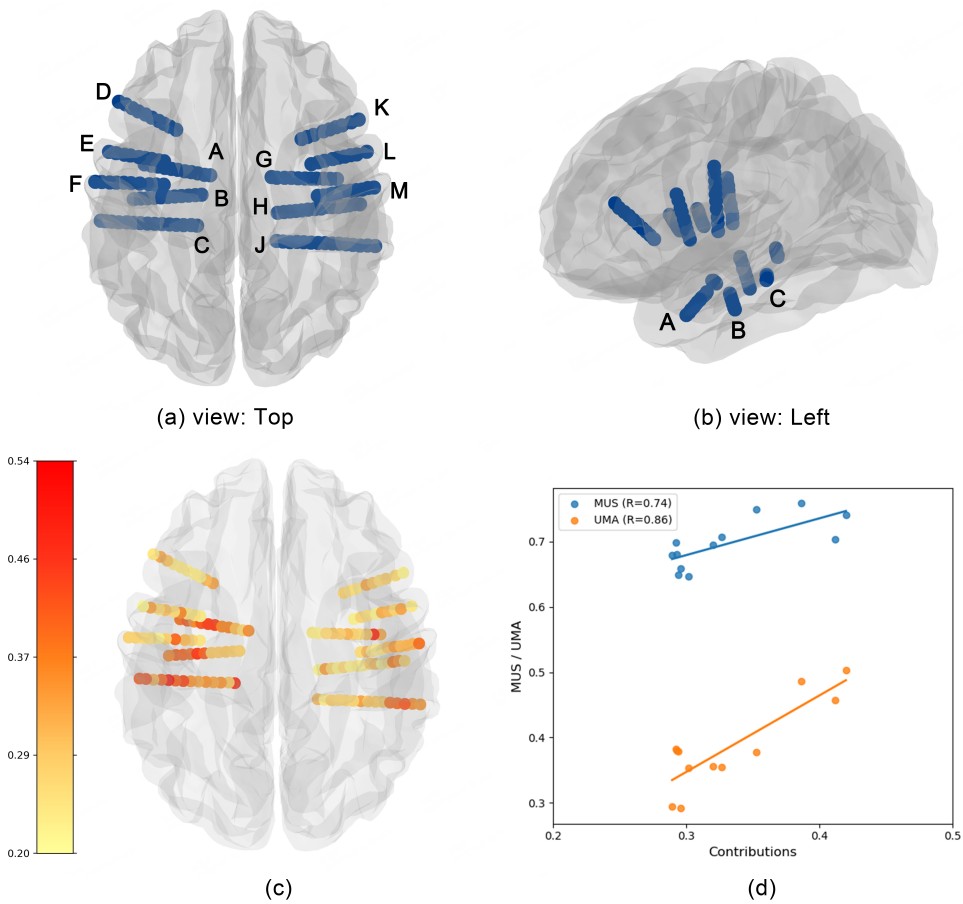

Figure 3: **Regional contribution analysis. (a,b)** Spatial distribution of electrodes and SEEG channels on the MNI template brain (top and left views). **(c)** Gradient-based channel saliency, with red indicating higher contribution.. **(d)** Positive correlations ($P \leq 0.01$) between electrode saliency and single-electrode decoding performance (UMA/MUS).

The spatial distribution of all implanted electrodes and recording channels used for decoding is shown on the standard Montreal Neurological Institute (MNI) template brain in Fig. 3 (a,b), from superior and left-lateral views. The coverage spans key temporal, insular, and limbic structures that are well established in the literature as supporting speech and semantic processing.

To quantify the contribution of different brain regions to semantic decoding, we employed two complementary analyses. First, we assessed how much semantic information can be extracted from individual recording sites in isolation. For each electrode, we trained and evaluated the decoder using only the corresponding SEEG channels as inputs, and measured decoding UMA and MUS. This provided an estimate of the semantic-informational content available locally at each site. Second, we examined how strongly each channel contributes to the model's decisions. We trained the model on the full multichannel SEEG input and then computed a gradient-based channel saliency measure (Simonyan et al., 2013; Schirrmeister et al., 2017). These saliency scores reflect the degree to which perturbing a given channel would affect the decoding loss, thereby quantifying its functional contribution within the learned decision function. The spatial distribution of saliency values is shown in Fig. 3 (c).

Each channel was subsequently mapped to its anatomical label using clinical annotations, enabling aggregation at both the channel and electrode levels. At the channel level, we observed particularly high saliency in the superior temporal gyrus and sulcus (STG/STS), which are core hubs of the

auditory-language pathway (Hickok & Poeppel, 2007; Friederici, 2011; Dronkers, 1996). At the electrode level, averaging saliency across channels from the same electrode identified a small set of electrodes with markedly elevated contributions, predominantly located in the superior and middle temporal cortices. Most high-contribution electrodes (A, B, C) were located in the left hemisphere, with several informative sites in the right temporal lobe. This is consistent with the well-established left-hemisphere dominance in semantic and speech processing (Hickok & Poeppel, 2007; Binder et al., 2009).

Finally, we examined the relationship between regional contribution and decoding performance. For each electrode, we computed the Pearson correlation between its average saliency value from the full-model analysis and its UMA and MUS scores from the electrode selection experiments. As shown in Fig. 3(d), saliency scores were significantly positively correlated with decoding accuracy for both metrics ($P \leq 0.01$), indicating that the anatomical regions most relied upon by the model during decoding are also those from which semantic information can be most effectively read out.

## J    BRAINMOSAIC IMPLEMENTATION

**EEG Encoder.**    The EEG Encoder processes raw EEG recordings into compact, context-rich neural representations. It consists of a ModernTCN (Luo & Wang, 2024) for low-level temporal modeling and a Transformer for global context integration and query-based representation. ModernTCN uses large and small convolutional kernels in a multi-stage architecture to capture both local patterns and long-range temporal dependencies, producing robust and noise-tolerant features. Given an input EEG sequence $x \in \mathbb{R}^{B \times C \times T}$, it outputs temporal tokens $X \in \mathbb{R}^{B \times N \times D}$, where $B$ is the batch size, $C$ is the number of channels, $T$ is the temporal length, and $N$ is the number of temporal patches. The Transformer combines these tokens with a fixed set of learnable queries to produce $K$ slot representations, each serving as a candidate semantic unit for the subsequent stages.

**Text Encoder.**    The text encoder in BRAINMOSAIC defines the open, continuous semantic space $\mathcal{V}$ using pre-trained embedding models and truncated to 256 dimensions. To build interpretable and stable semantic units, we expand each word into a short explanatory phrase via an LLM and embed it to reduce ambiguity and improve generalization. Sentences are embedded directly for global alignment, while high-level attributes are inferred by the LLM. For datasets without explicit word-level annotations, we apply simple tokenization, remove non-semantic function words, and embed the remaining content words. Similar embeddings are then clustered, merging variants such as "apple" and "apples" into a single unit. This process yields a stable, open-vocabulary semantic unit bank $U = u$ with embeddings $E(u)$, forming the foundation for alignment and retrieval.

---

**Algorithm 1** Training of BRAINMOSAIC

---

**Require:** Dataset $\mathcal{D} = \{(x_b, s_b)\}_{b=1}^B$; epochs $E$; weights $\lambda_{\text{cls}}, \lambda_{\text{attr}}, \lambda_{\text{global}}, \lambda_{\text{rep}}$; margin $m$.
**Ensure:** Final parameters $\theta$.
1: **for** $e = 1$ **to** $E$ **do**
2:     $\mathcal{L}_{\text{total}} \leftarrow 0$
3:     **for** each $(x_b, s_b) \in \mathcal{D}$ **do**
4:         $S_b, \{E(u_{b,i})\}_{i=1}^{n_b}, E(s_b), Z_b, U \leftarrow \text{TextEncoder}(s_b)$     *// text units, embeddings, global labels*
5:         $X_b \leftarrow \text{EEGEncoder}(x_b)$
6:         $\{\hat{y}_{b,j}, \hat{p}_{b,j}\}_{j=1}^K, \hat{s}_b, \{\hat{z}_b^{(c)}\}_{c=1}^C \leftarrow \text{SemanticRetriever}(X_b, Q)$
7:         $\hat{\sigma}_b \leftarrow \text{HungarianMatcher}(\{E(u_{b,i})\}, \{\hat{y}_{b,j}\})$
8:         $\mathcal{L}_{\text{Hungarian}}^{(b)} \leftarrow \sum_{j=1}^K \big[ t_{b,j} \cdot (1 - \text{sim}(E(u_{b,\hat{\imath}(j)}), \hat{y}_{b,\hat{\sigma}_b(j)})) + \lambda_{\text{cls}} \text{BCE}(t_{b,j}, \hat{p}_{b,\hat{\sigma}_b(j)}) \big]$
9:         $\mathcal{L}_{\text{global}}^{(b)} \leftarrow (1 - \text{sim}(E(s_b), \hat{s}_b)) + \lambda_{\text{attr}} \sum_{c=1}^C \text{CE}(z_b^{(c)}, \hat{z}_b^{(c)})$
10:         $\bar{\mathcal{I}}_b \leftarrow \{j : t_{b,j} = 0\}; \quad \mathcal{L}_{\text{rep}}^{(b)} \leftarrow \frac{1}{|\bar{\mathcal{I}}_b|} \sum_{j \in \bar{\mathcal{I}}_b} \frac{1}{|U|} \sum_{u \in U} \max(0, \text{sim}(\hat{y}_{b,j}, E(u)) - m)$
11:         $\mathcal{L}_{\text{retriever}}^{(b)} \leftarrow \mathcal{L}_{\text{Hungarian}}^{(b)} + \lambda_{\text{global}} \mathcal{L}_{\text{global}}^{(b)}$
12:         $\mathcal{L}_{\text{total}} \leftarrow \mathcal{L}_{\text{total}} + (\mathcal{L}_{\text{retriever}}^{(b)} + \lambda_{\text{rep}} \mathcal{L}_{\text{rep}}^{(b)})$
13:         **Backprop**$(\mathcal{L}_{\text{retriever}}^{(b)} + \lambda_{\text{rep}} \mathcal{L}_{\text{rep}}^{(b)})$; update partial $\theta$
14:     **end for**
15:     **EpochSummary:** optional logging with $\mathcal{L}_{\text{total}}$
16: **end for**

---

---

**Algorithm 2** Inference and Evaluation of BRAINMOSAIC

---

**Require:** Dataset $\mathcal{D} = \{(x_b, s_b)\}_{b=1}^B$; unit bank $U$ with embeddings $\{E(u)\}$;
  thresholds: existence $\delta$, cosine $\tau$; retrieval $k$; generation count $R = 5$.
**Ensure:** Per-sample predictions $\hat{S}_b$, $\hat{s}_b$, $\hat{T}_b$ and metrics $\text{UMA}_b$, $\text{MUS}_b$, $\text{SRS}_b$.

1: **for** each $(x_b, s_b) \in \mathcal{D}$ **do**
2:   $S_b$, $\{E(u_{b,i})\}_{i=1}^{n_b}$, $E(s_b)$, $Z_b \leftarrow \text{TextEncoder}(s_b)$
3:   $X_b \leftarrow \text{EEGEncoder}(x_b)$
4:   $\{\hat{y}_{b,j}, \hat{p}_{b,j}\}_{j=1}^K$, $\hat{s}_b$, $\{\hat{z}_b^{(c)}\}_{c=1}^C \leftarrow \text{SemanticRetriever}(X_b, Q)$
5:   $\hat{\sigma}_b \leftarrow \text{HungarianMatcher}\big(\{E(u_{b,i})\}_{i=1}^{n_b}, \{\hat{y}_{b,j}\}_{j=1}^K\big)$;   $t_{b,j} = \mathbb{1}[\exists i : \hat{\sigma}_b(i) = j]$
6:   $\hat{\mathbf{z}}_{b,i} \leftarrow \hat{y}_{b,\hat{\sigma}_b(i)}$ and $\mathbf{z}_{b,i}^* \leftarrow E(u_{b,i})$

$$\text{UMA}_b = \frac{1}{n_b} \sum_{i=1}^{n_b} \mathbb{I}\big(\text{sim}(\hat{\mathbf{z}}_{b,i}, \mathbf{z}_{b,i}^*) > \tau\big) \times 100\%, \qquad \text{MUS}_b = \frac{1}{n_b} \sum_{i=1}^{n_b} \text{sim}(\hat{\mathbf{z}}_{b,i}, \mathbf{z}_{b,i}^*)$$

7:   $\hat{S}_b = \{(u, p_{b,j}(u)) \mid \exists j,\ \hat{p}_{b,j} > \delta,\ u \in \text{Top-}k\ \text{NN}(\hat{y}_{b,j}, U)\}$
8:   $\text{Prompt}_b \leftarrow P(\hat{S}_b, \{\hat{z}_b^{(c)}\})$;   $\{\hat{T}_b^{(r)}\}_{r=1}^R \leftarrow G(\text{Prompt}_b)$
9:   $\text{SRS}_b = \frac{1}{R} \sum_{r=1}^R \text{sim}\big(E(\hat{T}_b^{(r)}), E(s_b)\big)$
10: **end for**

---

# K    RELATED WORK

## K.1    GENERIC LANGUAGE DECODING

### K.1.1    CONCEPT DECODING

Early research in intended decoding focused primarily on decoding short semantic units, such as individual words or simple phrases, from neural activity. Most approaches adopted closed-set classification paradigms, in which participants either imagined specific words or phonetics (Nieto et al., 2022; Lopez-Bernal et al., 2022), and processed externally presented stimuli in visual, auditory, or textual form (Wang et al., 2011; Liu et al., 2009; Vidal et al., 2010; Wilson et al., 2023). In some cases, these paradigms were followed by more complex linguistic tasks, such as picture naming or synonym matching in monolingual and bilingual conditions (Correia et al., 2015). Studies have revealed that the electrical potentials originating from the ventral temporal cortical surface in humans contain sufficient information to enable the spontaneous and near-instantaneous identification of a subject's perceptual state (Miller et al., 2016).

Despite recent advances, short-concept decoding still encounters significant hurdles. Existing studies generally focus on a limited number of highly distinct categories. Decoding tasks often revolve around isolated nouns instead of context-integrated expressions, and visual stimuli (Wang et al., 2011; Liu et al., 2009; Vidal et al., 2010; Sabra et al., 2020) are predominant in these research paradigms. These limitations severely restrict the real-world applicability of short-concept decoding. Since isolated concepts are difficult to combine into coherent expressions, the current approach struggles to meet the demands of actual language use.

### K.1.2    SENTENCE TOPIC DECODING

With the emergence of large language models (Zhao et al., 2023; Chang et al., 2024) and pre-trained models, the scope of research has broadened. Initially centered on emotion classification and motor imagery (MI) tasks, it has now extended to word and sentence decoding tasks (Zhang et al., 2025a). Moving from single concepts to sentence-level representation introduces new complexities in both neural modeling and linguistic reconstruction. Evidence from intracranial recordings and neuroimaging studies indicates that sentence processing relies on distributed and interactive cortical architectures rather than being confined to traditional language areas. Key regions such as the inferior frontal cortex and posterior temporal cortex play complementary roles in integrating semantic information, while the ventromedial prefrontal cortex is engaged during later stages to support high-level contextual interpretation and decision-making (Murphy et al., 2023).

In recent years, sentence-level decoding has attracted considerable attention. The DRYAD dataset (Broderick et al., 2018) records EEG data from participants listening to narrative speech, while ZuCo (Hollenstein et al., 2018) combines EEG and eye-tracking signals during reading. ZuCo 2.0 (Hollenstein et al., 2019) further expands the scale and diversity of participants. In the Chinese domain, ChineseEEG2 (Chen et al., 2025) collects EEG data during aloud reading and story listening, and Chisco (Zhang et al., 2024) focuses on both reading aloud and recall tasks. Although sentence decoding offers a more complete representation of language compared to single-word approaches, it also presents persistent challenges. Most methods are unable to reconstruct coherent sentences, relying instead on forced classification into predefined categories or extracting phrase-level embeddings from spontaneous speech, which fails to capture the diversity of natural linguistic contexts. The scarcity of large-scale, publicly available datasets further limits progress, while the integration of LLM-based models remains unstable and inconsistent in practice (Shams et al., 2025; Lu et al., 2025).

### K.1.3 SPEECH-BASED LANGUAGE DECODING

A parallel line of research has explored speech-based decoding strategies, which link the coordinated activity of vocal-tract articulatory and motor-planning cortical representations (Silva et al., 2024). These approaches typically require neural recordings obtained during overt speech production, which severely restricts their applicability to individuals with impaired articulation. Although recent work has demonstrated the feasibility of decoding during covert articulation, where participants silently attempt speech production (Moses et al., 2021), the resulting decoding speed remains suboptimal and performance falls short of real-time requirements. This limitation is particularly evident in non-alphabetic languages such as Chinese, which present additional challenges due to the complexity of logographic characters and tonal variations. Moreover, existing speech-motor-based models are largely confined to closed-set classification of syllables or tonal patterns during reading tasks (Liu et al., 2023; Zheng et al., 2024), offering limited generalization and low efficiency.

By contrast, semantic concepts in the brain constitute modality- and language-independent entities that underpin human cognition. Their distributed neural representation remains relatively consistent across linguistic systems, suggesting that decoding approaches centered on conceptual semantics, rather than phonological articulation, hold greater potential for restoring communication in individuals with speech and motor deficits.

### K.2 SET MATCHING

Set matching has emerged as a key paradigm in object detection and related recognition tasks. Early approaches were built on predefined anchors or centers, where models predicted targets relative to initial guesses (Zhang et al., 2020). However, their performance was highly sensitive to the design of these priors. Subsequent efforts introduced set-based losses (Redmon et al., 2016), most notably through the use of the Hungarian algorithm (Kuhn, 1955), which establishes a one-to-one assignment between predictions and ground-truth objects. However, such methods still relied on contextual cues, such as bounding box coordinates, and embedded prior knowledge into the pipeline. More recent research explored recurrent detectors that directly predict object sets through encoder-decoder architectures combined with bipartite matching losses, paving the way for the fully end-to-end Detection Transformer (DETR) (Carion et al., 2020). The development of DETR further advanced the set matching framework by demonstrating the feasibility of casting detection as a direct set prediction problem. More broadly, recent work has addressed the lack of positive samples and accelerate convergence (Dai et al., 2021), through strategies such as data augmentation (Fang et al., 2024) and one-to-many training (Jia et al., 2023), which enhance matching accuracy and improve model robustness. Inspired by DERT, we adopt set matching to treat EEG decoding as a structured prediction problem.

