# OpenReview forum: "Assembling the Mind's Mosaic: Towards EEG Semantic Intent Decoding"
_ICLR.cc/2026/Conference — ICLR 2026 Poster_

### Official Review · Reviewer_y3DJ · 2025-10-30

**Soundness:** 3
**Presentation:** 3
**Contribution:** 3
**Rating:** 8
**Confidence:** 4

**Summary:**

The paper proposes Semantic Intent Decoding (SID) and a concrete model, BRAINMOSAIC, for translating EEG or SEEG into natural language by first decoding a variable set of semantic units, aligning them in a continuous text embedding space, and then reconstructing a sentence with an LLM constrained by those units. The pipeline comprises a set-matching Semantic Decomposer trained with a Hungarian loss, a Semantic Retriever that aligns slot embeddings to word and sentence embeddings and predicts global attributes, and a Semantic Decoder that prompts an LLM to generate text from retrieved units. The work targets semantic decoding during reading, listening, or imagined speech across Chinese and English datasets, plus a private SEEG dataset, and claims improvements over classification and unconstrained generation using embedding based metrics UMA, MUS, and SRS

**Strengths:**

1. The three principles (compositionality, continuity, fidelity) are well-motivated by linguistic and neuroscience evidence.
2. Interpretable pipeline: slots -> ranked retrieval -> prompted gen beats blackbox E2E.
3. Consistent empirical gains across multiple datasets and baselines with both concept level and sentence level metrics.
4. Comprehensive comparison with other relevant baselines.
5. Extensive supplementary material includes dataset details, baseline descriptions, and sensitivity analyses.

**Weaknesses:**

1. Lack of qualitative examples of reconstruction quality. Without concrete examples, readers cannot verify whether predicted semantic unit sets are genuinely interpretable or noisy/scattered, (b) how well do the quantitative metrics match real semantic correctness. This falls below the standards in neuro-decoding literature where it's common to show a few samples of the proposed model's input-output vs baseline
2. Lack of comparability to standard text metrics. While semantics-first metrics are appropriate here, it's also important to have surface level metrics common in the literature such as BLEU score, WER for surface overlap, and BERTScore for semantic similarity.

**Questions:**

1. For continuous corpora, how did you prevent leakage between train and test when the same long passage is segmented into sentences?
2. Can you report error bars and conduct significance tests comparing BrainMosaic to baselines?
3. How sensitive are results to the choice of embedding model? What happens with random embeddings as an ablation?
4. For multi-subject datasets, what is cross subject variance? Can models trained on one subject decode another subject’s neural activity? How good is the generalizability
5. Did you perform any electrode selection or analyze which channels/regions contribute most to semantic unit prediction?
6. Table 5A shows declining UMA with vocabulary expansion. At what vocabulary size does performance approach random? Can you characterize this scaling law?

---

> ### Author Response · Authors · 2025-11-21
> **Reply to Reviewer y3DJ (1/3)**
>
> Thank you very much for recognizing our work.
>
> ###  W1&W2. Metrics and qualitative examples
>
> We appreciate the valuable suggestion to bridge the gap between quantitative metrics and qualitative interpretation.
> We address these points by integrating a unified evaluation of surface-level and semantic-level metrics and we respectfully invite the reviewer to examine **Figure 2 (Appendix D)**.
>
> **Inclusion of qualitative examples and reconstruction visualization.**
> We add Appendix D, including actual case studies from the datasets.
> These examples display the Ground Truth, our model's predictions, and baseline outputs side-by-side.
> This also allows for a direct verification of the interpretability of the predicted units and demonstrates how well the metrics align with human-perceived semantic correctness.
>
> **Incorporation of standard text metrics and metric selection.**
> - We formally adopt **BERTScore-F1** as a primary evaluation metric and update the main table to facilitate standardized comparison. However, as illustrated in Figure 2, we observe that BERTScore-F1 exhibits insufficient discrimination across the test samples. Consequently, while we include BERTScore-F1 as a standard benchmark for literature comparison, we continue to utilize SRS as the primary evaluation metric under the SID framework.
> - BLEU/WER: We also evaluate on surface-level metrics (BLEU-1, BLEU-2, BLEU-4, WER) and detailed these results in Appendix D.
>
> The qualitative examples illustrate specific cases where the model successfully recovers the semantic intention but uses different phrasing.
> This offer intuitive evidence of why surface metrics often misalign with the goals of the SID framework.
>
> ### Q1. Data leakage
>
> **Clarification on potential leakage in continuous corpora.**
> We verify that the sequential nature of the source text does not introduce leakage in our sentence-level decoding tasks, based on the following analyses:
> **a) Semantic discontinuity in adjacent sentences.**
> The semantic content often shifts significantly between adjacent sentences.
> Example from ChineseEEG-2 (translation in English):
>
> *"So the grown-ups were very pleased"*
>
> *"to have met such a sensible person"*
>
> *"I lived my life alone"*
>
> *"without anyone I could really talk to"*
>
> *"until six years ago"*
>
> As shown, they represent distinct semantic events.
>
> **b) Statistical verification of semantic independence.**
> We compare the semantic similarity of Adjacent pairs (sentences N and N+1) against Distant pairs (randomly sampled pairs with a gap > 10).
> We calculate the embedding similarity for both groups and perform an independent two-sample t-test.
> As shown in the table below, we found no significant difference in semantic similarity between adjacent and distant sentences across all datasets.
>
> | Dataset       | t-value | p-value |
> |---------------|---------|---------|
> | ZuCo-SR       | 0.7810  | 0.4451  |
> | ZuCo-NR       | 1.4690  | 0.1666  |
> | ZuCo-TSR      | 0.6061  | 0.5524  |
> | ChineseEEG-2  | 1.5266  | 0.1738  |
>
> **Differentiation between leakage and generalization.**
> From a neural perspective, even if a subject reads two semantically similar sentences, their corresponding EEG representations often exhibit distinct patterns due to temporal variability and neural noise.
> If the model successfully maps distinct EEG patterns to similar semantic representations, this constitutes semantic generalization, not data leakage.
> We strongly agree with the reviewer that modeling long-context semantic coherence is a valuable insight.
> We acknowledge that incorporating paragraph-level context is an important direction for future clinical experiments and research.
>
> ### Q2. Error bars and significance tests
>
> We update Table 3 to include both error bars (standard deviations) and significance test results.
> The analysis confirms that BrainMosaic demonstrates a statistically significant improvement (p < 0.001) over the best baselines across all metrics and datasets.

---

> > ### Author Response · Authors · 2025-11-21
> > **Reply to Reviewer y3DJ (2/3)**
> >
> > ### Q3. Choice of embedding model
> >
> > We appreciate this insightful question.
> > We add a systematic analysis in Appendix E, comparing different embedding models, construction strategies, and a random baseline.
> >
> > **Sensitivity to Embedding Models and Strategies.**
> > We compare two embedding models (Qwen and Doubao) using two different construction strategies:
> > - **Word-only**: Generating embeddings directly from the word itself (e.g., "orchard").
> > - **Explanation-augmented**: Generating embeddings from the word followed by its dictionary definition (e.g., "orchard: an intentional plantation of trees...").
> >
> > As shown below, the Doubao (Explanation) configuration yields the best performance.
> > Notably, the "Explanation" strategy consistently outperforms the "Word-only" strategy, suggesting that richer semantic descriptions help construct a more discriminative continuous semantic space.
> >
> > **Impact of Random Embeddings.**
> > We construct a random baseline using fixed 256-dimensional random vectors (similarity distribution $\in (0, 1)$, $\mu \approx 0.5$, $\sigma \approx 0.1$) assigned to each semantic unit.
> > The use of random embeddings caused a drastic degradation in performance across all metrics. Crucially, the predicted sets and sentences indicate that effective predictions are limited solely to high-frequency units, reflecting a fallback to corpus priors rather than valid neural decoding.
> >
> > **Conclusion:** These results underscore that the quality of the embedding space is one of the critical determinants of decoding performance. Furthermore, this implies a promising scaling property for our framework: as LLM embedding technologies continue to advance in expressiveness and granularity, our method will naturally inherit these improvements, potentially leading to further gains in decoding accuracy without architectural changes.
> >
> > |                      | Clinical |        |        |         | Chisco |        |        |         |
> > | -------------------- | -------- | ------ | ------ | ------- | ------ | ------ | ------ | ------- |
> > |                      | UMA      | MUS    | SRS    | BERT-F1 | UMA    | MUS    | SRS    | BERT-F1 |
> > | random               | 0.0644   | 0.6962 | 0.5067 | 0.5962  | 0.0417 | 0.6882 | 0.4932 | 0.5719  |
> > | qwen (word)          | 0.1994   | 0.7194 | 0.6014 | 0.6166  | 0.1601 | 0.7073 | 0.6044 | 0.5873  |
> > | qwen (explanation)   | 0.2082   | 0.7389 | 0.6271 | 0.6249  | 0.1685 | 0.7108 | 0.6121 | 0.5904  |
> > | doubao (word)        | 0.6373   | 0.8014 | 0.6578 | 0.6480  | 0.5502 | 0.7986 | 0.6098 | 0.6022  |
> > | doubao (explanation) | 0.6596   | 0.8124 | 0.6651 | 0.6629  | 0.5617 | 0.8009 | 0.6206 | 0.6195  |
> >
> > ### Q4. Cross subject generalizability
> >
> > We follow the mainstream protocols in non-invasive neural language decoding, where models are trained and tested independently for each subject [1-2].
> > We did not conduct direct cross-subject decoding experiments for the following theoretical and physiological reasons:
> >
> > - **Anatomical Heterogeneity**: High-level semantic decoding involves complex multi-region coordination. Significant anatomical variations in brain structure and neuronal distribution across individuals make the direct transfer of decoding models highly challenging [3].
> > - **Individualized Semantic Representation**: Neural representations for identical semantic concepts vary significantly between subjects. These differences are driven by distinct physiological structures, linguistic experiences, and cognitive processing patterns [4-5].
> > - **Complexity of Fine-grained Decoding**: Unlike classification tasks with a limited number of discrete categories (e.g., sleep staging or motor imagery) where cross-subject transfer is more feasible, our task involves high-dimensional, fine-grained semantic decoding. In such a vast semantic space, the requirement for precise feature alignment is exponentially higher, making the model extremely sensitive to inter-subject variability. Consequently, the within-subject setting remains the universal standard for current open-vocabulary decoding research.
> >
> > While we acknowledge that cross-subject modeling is a critical goal for the BCI community, it remains a distinct challenge.
> >
> >
> > [1] Feng, Chen, et al. "Acoustic inspired brain-to-sentence decoder for logosyllabic language."
> >
> > [2] Murad, Saydul Akbar, and Nick Rahimi. "Unveiling thoughts: A review of advancements in eeg brain signal decoding into text."
> >
> > [3] Orban, Mostafa, et al. "A review of brain activity and EEG-based brain–computer interfaces for rehabilitation application."
> >
> > [4] Liu, Xin, et al. "Individual differences in the neural architecture in semantic processing."
> >
> > [5] Ibáñez, Agustín, et al. "Ecological meanings: A consensus paper on individual differences and contextual influences in embodied language."

---

> > ### Author Response · Authors · 2025-11-21
> > **Reply to Reviewer y3DJ (3/3)**
> >
> > ### Q5. Channel/region contribution
> >
> > We add a comprehensive analysis to identify the key brain regions driving the decoding performance.
> > We respectfully direct the reviewer to **Appendix I (and Figure 11)** for the visualization and detailed discussion.
> >
> > We employ two complementary approaches:
> > (1) Gradient-based Saliency Analysis to determine functional contribution, and (2) Single-Electrode Decoding to measure the intrinsic contribution.
> >
> > To summarize our findings:
> > The analysis reveals that the highest-contributing electrodes are predominantly concentrated in the temporal lobe, exhibiting a marked Left-Hemisphere Dominance.
> > This also aligns with existing findings [6-8].
> >
> >
> > ### Q6.Scalability
> >
> > We provide a detailed analysis of the scalability in **Global Comment 2**.
> >
> > We clarify that performance does not approach random guessing as vocabulary expands.
> > As the vocabulary expands, new words (e.g., synonyms, derivatives) naturally map into existing regions rather than creating distinct new conceptual dimensions.
> > Furthermore, linguistic evidence suggests that the effective human semantic space is finite. Therefore, performance stabilizes rather than degrading continuously.
> >
> >
> > [6] Hickok, Gregory, and David Poeppel. "The cortical organization of speech processing."
> >
> > [7] Friederici, Angela D. "The brain basis of language processing: from structure to function."
> >
> > [8] Binder, Jeffrey R., et al. "Where is the semantic system? A critical review and meta-analysis of 120 functional neuroimaging studies."

---

### Official Review · Reviewer_xT72 · 2025-10-31

**Soundness:** 2
**Presentation:** 2
**Contribution:** 3
**Rating:** 4
**Confidence:** 4

**Summary:**

The paper introduces Semantic Intent Decoding (SID), a novel framework for brain-to-language translation that models communicative intent as a set of compositional semantic units rather than relying on fixed labels or unconstrained generation.

**Strengths:**

The paper proposes an intriguing motivation and introduces a novel perspective for brain-to-text decoding. By modeling intent as an unordered and variable set of semantic units, it moves beyond traditional fixed-label or sequential decoding paradigms, potentially offering a more brain-plausible representation of semantic processing.

**Weaknesses:**

1.Lack of robustness evaluation under input noise: the paper does not assess model performance under noisy or corrupted EEG inputs, which weakens the significance of the results reported in Tables 3, 4, and 5. Given the inherent noisiness of EEG signals, such evaluations are essential to validate the practical utility of the proposed method.

2.Insufficient justification for LLM-based sentence generation: the use of LLMs for sentence reconstruction raises concerns about data contamination, especially if the test set sentences or similar phrasings were present in the LLM’s pretraining corpus. The authors provide no strong evidence (e.g., n-gram overlap analysis or controlled LLM ablation) to rule out this possibility, which undermines the credibility of the generation results.

3.Missing comparison with standard generation metrics and SOTA methods: the paper does not compare with mainstream brain-to-text decoding methods using standard generation metrics such as WER, CER, BLEU, ROUGE, METEOR, or BERTScore. This omission makes it difficult to benchmark the proposed method against existing literature and assess its true advancement in the field.

**Questions:**

1.How was K (number of semantic slots) chosen per dataset? Was it tuned? How does performance vary with K?

2.Would the model scale to larger vocabularies? What are the computational bottlenecks?

---

> ### Author Response · Authors · 2025-11-21
> **Reply to Reviewer xT72 (1/2)**
>
> We sincerely thank the reviewer for the valuable comments.
>
> ### W1. Robustness evaluation under noise.
>
> **We clarify the intrinsic noise naturally present in our data.** Nevertheless, to fully address the suggestion, we conduct additional synthetic noise experiments.
>
> **EEG signals contain characteristically noise.** The EEG data used in our study naturally contains significant noise from environmental interference (acquisition equipment) and intrinsic brain state fluctuations (caused by the subject's alertness, attention, micro-movements, and random physiological/psychological processes) [1-2]. This is particularly evident in language and cognitive tasks, where neural signals can vary drastically even when the same subject listens/reads the exact same sentence twice. Consequently, the datasets already encompass a rich variety of real-world noise, implicitly testing the model's robustness.
>
> **Evaluation are typically conducted on naturally noisy data,** as the real-world variability is already substantial. Representative works typically evaluate models directly on naturally noisy data rather than synthetic noises [3-5].
>
> **Experiments with synthetic noises.** We perform offline noise interference tests on the Clinical dataset with 3 common types of noise, each type was applied at 3 intensity levels. Considering the inherently high SNR of intracranial recordings [6], the severe level condition represents a worst-case scenario that is rarely encountered in standard data acquisition.
>
> - **Gaussian White Noise:** Simulating noise from unstable electrode-sensor contact;
> - **50Hz Line Noise:** Simulating the most common power frequency interference from acquisition equipment;
> - **Baseline Drift:** Simulating slow changes in electrode impedance or shifts in the subject's state.
>
> **Results show the robustness** **of our model.** The model demonstrates remarkable stability under mild to moderate noise corruption, with performance fluctuations remaining within 2% relative to the original results.
>
> **Table: Model Performance under Different Noise Conditions**
>
> |         | Orig   |   Gaussian     |  |        |    50Hz Line     | |        |    Drift    |   |        |
> | ------- | ------ | :----: | :------: | :----: | :----: | :-------: | :----: | :----: | :----: | :----: |
> |         |        |  40dB  |   20dB   |  10dB  | 0.1std |  0.3std   | 0.5std | 0.1std | 0.3std | 0.5std |
> | **UMA** | 0.6596 | 0.6539 |  0.6333  | 0.6170 | 0.6397 |  0.6226   | 0.5885 | 0.6326 | 0.6277 | 0.6083 |
> | **MUS** | 0.8124 | 0.8104 |  0.8075  | 0.8069 | 0.8053 |  0.8060   | 0.7962 | 0.8088 | 0.8063 | 0.8044 |
>
> ### W2. Unverified LLM Contamination
>
>
> **We would like to clarify the fundamental premise of our framework and provide evidence that LLM contamination is not a valid concern in our experimental setup.**
>
>
>
> **a) Our research objective is** **cognitive decoding** (mapping EEG signals to semantic intents), not enhancing LLM generation capabilities using EEG. As explicitly stated in **Introduction-3 (Fidelity)** and **Section 2.3**, the LLM acts as an auxiliary tool to translate the decoded sets into fluent, grammatical sentences.
>
>
>
> **b) We employ frozen LLMs without any fine-tuning on our corpus.** We utilize standard LLMs directly. We strictly do not fine-tune these models on our text corpus. While modern LLMs have indeed been exposed to vast amounts of data during pre-training, their role in our framework is limited to syntactic assembly based on provided constraints.
>
>
>
> **c) The semantic content is derived exclusively from EEG signals.** As detailed in **Section 2.4**, our training process focuses entirely on mapping EEG signals to the semantic representation space. The LLM is completely excluded from the training phase. According to Principle 3, the LLM's function is to faithfully reconstruct sentences. Therefore, the semantic intent of the generated sentence is **determined** by the EEG signal, not by the LLM's prior knowledge.
>
>
>
> **d) Experiment results:** Control experiments with random inputs confirm that the LLM cannot hallucinate correct sentences without accurate EEG decoding (see **Global Comments 1**). This directly proves that the performance of our model stems from the accurate decoding of EEG signals.
>
>
> [1] Bergmann, Til O. "Brain state-dependent brain stimulation."
>
> [2] Liu, Xin, et al. "Individual differences in the neural architecture in semantic processing."
>
> [3] Zheng, Hui, et al. "Du-IN: Discrete units-guided mask modeling for decoding speech from Intracranial Neural signals."
>
> [4] Feng, Chen, et al. "Acoustic inspired brain-to-sentence decoder for logosyllabic language."
>
> [5] Zhang, Zihan, et al. "Chisco: An EEG-based BCI dataset for decoding of imagined speech."
>
> [6] Parvizi, Josef, and Sabine Kastner. "Promises and limitations of human intracranial electroencephalography."

---

> > ### Author Response · Authors · 2025-11-21
> > **Reply to Reviewer xT72 (2/2)**
> >
> > ### W3. Standard Generation Metrics and Baseline Comparison
> >
> > **Inclusion of standard metrics.** We supplement our analysis with traditional generation metrics. These results and a corresponding analysis can be found in **Global Comment 3** and **Appendix D**.
> >
> > **Justification for the limitations of standard metrics in SID.** While we add the metrics for completeness, we emphasize (as detailed in Appendix D) that standard metrics do not fully align with the SID framework. Our work is motivated by the need to maximize the restoration of semantic intention from EEG signals, which is a conceptual and abstract decoding task. Consequently, our proposed semantic metrics offer a more accurate assessment, whereas standard metrics face challanges like penalizing semantically equivalent but phrased-differently outputs.
> >
> > **Clarification on existing baseline:** Regarding the concern that the paper misses comparisons with mainstream methods, we respectfully direct the reviewer's attention to Table 3 in the main text. It is also important to note that our work establishes a brand-new paradigm (SID), which standard classification-based models cannot natively follow. Despite this structural incompatibility, we adapt these mainstream methods into modified versions, enabling us to provide the rigorous comparison presented in the table.
> >
> > ### Q1. Systematic analysis of hyper-parameter K
> >
> > We add a comprehensive experiment regarding the number of semantic slots (K), details in Appendix G, testing a comprehensive range of candidate values.
> > To understand slot utilization, we monitor the Number of Unused Slots (NUS), which indicates the average count of slots where the model predicts an "no-object" during testing.
> >
> > **Results:** The results below demonstrate that performance is limited only when K is insufficient. As K increases from 10 to 50, all the key metrics exhibit minimal variance, with performance fluctuations remaining below 2% relative to the peak, while the NUS rises proportionally. This indicates that the model is robust to the choice of K, provided it meets a minimum semantic threshold.
> >
> > **Table: Model Performance under Different K values**
> >
> > |       | Clinical |         |         |         | Chisco  |         |         |         |
> > | ----- | :------: | :-----: | :-----: | :-----: | :-----: | :-----: | :-----: | :-----: |
> > | **K** | **NUS**  | **UMA** | **MUS** | **SRS** | **NUS** | **UMA** | **MUS** | **SRS** |
> > | 1     |   0.00   | 0.1493  | 0.8023  | 0.5123  |  0.00   | 0.0907  | 0.7503  | 0.4988  |
> > | 2     |   0.00   | 0.3433  | 0.8208  | 0.5891  |  0.00   | 0.2706  | 0.7921  | 0.5772  |
> > | 5     |   0.61   | 0.6290  | 0.8063  | 0.6499  |  1.59   | 0.5299  | 0.7977  | 0.5937  |
> > | 10    |   5.36   | 0.6596  | 0.8124  | 0.6651  |  5.75   | 0.5617  | 0.8009  | 0.6206  |
> > | 15    |  10.35   | 0.6559  | 0.8162  | 0.6624  |  11.38  | 0.5590  | 0.7989  | 0.6172  |
> > | 25    |  20.64   | 0.6418  | 0.8104  | 0.6683  |  21.23  | 0.5577  | 0.8012  | 0.6208  |
> > | 50    |  45.56   | 0.6482  | 0.8138  | 0.6636  |  45.76  | 0.5601  | 0.7954  | 0.6160  |
> >
> > ### Q2. Scalability and computational bottlenecks
> >
> > **Existing experiments and theoretical rationale confirm scalability.** Regarding the question of whether the model scales to larger vocabularies, we respectfully refer the reviewer to **Section 3.3** of the main text. We systematically conduct experiments on both vocabulary expansion and dataset expansion to validate scalability. Furthermore, we supplement **Global Comment 2** with the rationale underlying our scalability experimental setup.
> >
> >
> > **Mature retrieval algorithms effectively resolve computational bottlenecks.** The backbone of our model remains lightweight during the forward pass. The only component that computationally scales with vocabulary size is the retrieval process. However, large-scale vector retrieval is a well-established and mature field in the NLP and LLM communities [7-9]. Consequently, even if future EEG semantic resources expand significantly, we can directly employ efficient ANN algorithms to resolve any potential latency issues effectively.
> > Besides, existing researches indicate that approximately 12,000+ cue words are sufficient to cover the human semantic space [10], while our configuration (30,000 units) already exceeds this threshold, ensuring that the semantic space is saturated and robust.
> >
> > [7] Johnson, Jeff, Matthijs Douze, and Hervé Jégou. "Billion-scale similarity search with GPUs."
> >
> > [8] Malkov, Yu A., and Dmitry A. Yashunin. "Efficient and robust approximate nearest neighbor search using hierarchical navigable small world graphs."
> >
> > [9] Aumüller, Martin, Erik Bernhardsson, and Alexander Faithfull. "ANN-Benchmarks: A benchmarking tool for approximate nearest neighbor algorithms."
> >
> > [10] De Deyne, Simon, et al. "The “Small World of Words” English word association norms for over 12,000 cue words."

---

### Official Review · Reviewer_MrKn · 2025-11-01

**Soundness:** 3
**Presentation:** 4
**Contribution:** 3
**Rating:** 6
**Confidence:** 4

**Summary:**

The paper proposes to view EEG/SEEG-to-language decoding as predicting a variable-size, order-invariant set of semantic units (SID), rather than fixed-class decoding or fully free-form generation. It instantiates this with BRAINMOSAIC, including i) an EEG encoder + query slots, ii) a semantic retriever that aligns slots to an open-vocabulary unit bank, and iii) an LLM that does semantics-constrained generation from those units. Experiments on three public EEG datasets (Chinese + English) and one private SEEG dataset, using UMA, MUS, and sentence-level similarity, aim to show: (i) set-style decoding is reasonable, (ii) continuous semantic space helps scalability, (iii) constrained generation improves fidelity.

**Strengths:**

1. Interesting formulation: treating an utterance as a set of semantic units is a neat and fairly novel way to handle variable-length EEG semantics.

2. Well-structured method: the three design principles (compositionality, continuity/expandability, fidelity) map cleanly to three modules.

3. Reasonable experiment designs: the same idea is run on Chinese imagined speech, Chinese naturalistic reading, English reading, and a clinical SEEG case, which supports the claim that the approach is not tied to a single dataset.

**Weaknesses:**

1. **Train/validation split is underspecified:** the paper only mentions a unified 8:2 train–test split, but does not say whether this is by subject or by trial. Can the model guess which sentence it is by the sample length?

2. **No true random / text-prior baselines**: the main metrics (UMA, MUS, sentence similarity) are not compared against (i) picking the same number of units at random or (ii) a text-only/corpus-frequency prior. This makes it hard to see how much of the score actually comes from EEG, especially since the method later calls an LLM. The paper reports `MUS_exp` but doesn’t clearly define it.

3. Experiments don’t cleanly isolate the three research questions: the section is organized around “set is better,” “continuous space scales,” and “constrained generation is more faithful”. Several experiments change multiple components at once (set + LLM + thresholds), so it’s hard to attribute the improvements to the claimed factor.

**Questions:**

1.Is the 8:2 split done by subject or by trial on each dataset? Is there possibility of data leakage?
2.Please add complete random baselins for UMA/MUS/SRS. Please describe the MUS_exp more clearly.
3.Please add ablations where only one of the three design choices (set, continuous space, LLM-constrained decoding) is changed at a time, so we can see which part actually drives the gains?
4.Please distinguish clearly between ChineseEEG and ChineseEEG-2 as they are two distinct datasets.
Some references are missing. For example, there should be references for this statement: “Alternatively, a more recent direction seeks to enhance expressive capacity by mapping neural signals directly into the latent representation space of large language models (LLMs).”

---

> ### Author Response · Authors · 2025-11-21
> **Reply to Reviewer MrKn**
>
> Thank you very much for recognizing our work.
>
> ### W1&Q1. Data splitting and leakage
>
> **Clarification on the in-subject data splitting strategy.**
>
> We thank the reviewer for identifying these details regarding the experimental setup.
> We update **Section 3.1** and **Appendix A** to explicitly state that all experiments were conducted under an in-subject setting.
> We follow the standard protocol in language neural-decoding by performing independent training-testing for each subject (an 8:2 split by trial).
> This is essential for two reasons: first, significant anatomical variations make cross-subject transfer inherently challenging; second, unlike limited-category classification tasks, our work involves high-dimensional, fine-grained semantic decoding. Therefore, the requirement for precise feature alignment is exponentially higher, making the model more sensitive to inter-subject variability.
>
> **The model does not rely on sample length to infer semantics.**
> - Trial Independence: Each subject reads/listens to the corpus once. If a subject participates in both tasks, they are treated as two independent "subjects", ensuring no overlap between training and testing.
> - Variable-Length Datasets: For ChineseEEG-2 and ZuCo, we perform a statistical hypothesis test to check if EEG duration correlates with semantic content. We calculate the Pearson correlation coefficient between the absolute difference in segment lengths and semantic similarity of the corresponding sentence labels.
>
> As shown below, the correlations are negligible with non-significant p-values (p > 0.05). This statistically confirms that segment length contains no predictive information regarding semantic similarity.
>
> | Dataset             | r       | p      |
> | ------------------- | ------- | ------ |
> | ChineseEEG-2-read   | -0.0058 | 0.6804 |
> | ChineseEEG-2-listen | -0.0071 | 0.6176 |
> | ZuCo-SR             | -0.0184 | 0.1937 |
> | ZuCo-NR             | -0.0222 | 0.1169 |
> | ZuCo-TSR            | -0.0086 | 0.5424 |
>
> ### W2&Q2. Random / text-prior baselines
>
> We agree that incorporating random and text-prior baselines is crucial to decouple the contribution of the EEG decoding module.
>
> **Inclusion of random and text-prior baselines.**
> Detailed analyses are in **Global Comment 1** and **Appendix C.5**.
> These results confirm that our method significantly outperforms random and frequency-based priors.
>
> **Clarification of $MUS_{exp}$.**
> $MUS_{exp}$ serves as a baseline reflecting the intrinsic semantic density of the corpus.
> It is calculated as the average embedding similarity between two semantic units drawn uniformly and independently from the corpus without frequency distributions.
> Conceptually, it represents the theoretical lower bound of MUS when both the labels and predictions are random sets.
>
> ### W3&Q3. Ablation experiments
>
> We agree that isolating the contribution of each component is essential. We add the ablation study in **Section 3.5** and **Table 5**:
>
> - **w/o Set.** We replace the unit-level set objective with a sentence-level alignment objective. The EEG encoder is trained to align directly with the sentence embedding. During inference, the Top-K semantic units closest to the predicted embedding are selected.
> - **w/o Continuous Space.** We replace the semantic retrieval module with a multi-label classification model. Top-K predicted labels are treated as the decoded semantic units.
> - **w/o LLM.** Using the semantic units decoded by the standard pipeline, we concatenate the words in random orders (repeated 50 times) and select the sequence with the highest SRS.
>
> **Table: Ablation results of BrainMosaic**
>
> |                   |  Clinical  |            |            |      Chisco      |      |            |
> | ----------------- | :--------: | :--------: | :--------: | ---------- | :--------: | :--------: |
> | Model Variant |    UMA     |    MUS     |    SRS         |    UMA     |    MUS     |    SRS     |
> | w/o Set      |   0.0792   |   0.7052   |   0.5721        |   0.0260   |   0.6690   |   0.5506   |
> | w/o ContSpace |    0.0137  |  0.6393    |   0.4604       |   0.0044   |  0.6009    |   0.3727   |
> | w/o LLM       | **0.6596** | **0.8124** |   0.5456       | **0.5617** | **0.8009** |   0.5133   |
> | Full    | **0.6596** | **0.8124** | **0.6651**  | **0.5617** | **0.8009** | **0.6206** |
>
> ### Q4. Dataset usage and missing references
>
> **Clarification on ChineseEEG-2 usage.** We apologize for the typo. All experiments are conducted using ChineseEEG-2. The original version is excluded because it lacks the necessary granularity for semantic segmentation. We correct the labeling in the revised manuscript.
>
> **Inclusion of missing references.** We add citations regarding End-to-End Generation in Section 1.

---

### Author Response · Authors · 2025-11-21
**Global Comment (1/3)**

### 1. Random & Text-Prior Baselines

Suggested by Reviewer MrKn, we implement two levels of baselines to decouple EEG contributions from corpus statistics (Appendix C.5). All random results represent the average of 3 independent runs over the entire test set.

***1. Label-level***
Keeps the prediction pipeline fixed but compares against random concept and sentence labels.

***2. Prediction-level***
Tests statistical guessing against truth labels by bypassing the EEG encoder.
- Random-prior: Uniform random selection.
- TopK-prior: Deterministic Top-K frequent units.
- Freq-prior: Frequency-weighted random selection.

As shown below, BrainMosaic significantly outperforms all random and text-prior baselines across all metrics. This consistent superiority confirms that the performance stems from valid neural decoding rather than statistical guessing.


|               |  Clinical  |            |            |   Chisco   |            |            | ChineseEEG-2 |            |            |
| ------------- | :--------: | :--------: | :--------: | :--------: | :--------: | :--------: | :----------: | :--------: | :--------: |
|               |    UMA     |    MUS     |    SRS     |    UMA     |    MUS     |    SRS     |     UMA      |    MUS     |    SRS     |
| Label Random  |   0.2472   |   0.6878   |   0.5394   |   0.1390   |   0.6987   |   0.4997   |    0.1308    |   0.6955   |   0.4683   |
| Random-prior  |   0.0563   |   0.6392   |   0.5056   |   0.0152   |   0.6185   |   0.4924   |    0.0112    |   0.5815   |   0.5129   |
| Topk-prior    |   0.1510   |   0.6884   |   0.5418   |   0.1215   |   0.6793   |   0.5465   |    0.1159    |   0.6606   |   0.5282   |
| Freq-prior    |   0.1008   |   0.6540   |   0.5321   |   0.0289   |   0.6365   |   0.5124   |    0.0332    |   0.6046   |   0.5170   |
| BrainMosaic | **0.6596** | **0.8124** | **0.6651** | **0.5617** | **0.8009** | **0.6206** |  **0.3707**  | **0.7687** | **0.6163** |

|               |   ZuCoSR   |            |            |   ZuCoNR   |            |            |  ZuCoTSR   |            |            |
| ------------- | :--------: | :--------: | :--------: | :--------: | :--------: | :--------: | :--------: | :--------: | :--------: |
|               |    UMA     |    MUS     |    SRS     |    UMA     |    MUS     |    SRS     |    UMA     |    MUS     |    SRS     |
| Label Random  |   0.2557   |   0.7591   |   0.5499   |   0.2640   |   0.7339   |   0.5297   |   0.2588   |   0.7084   |   0.5009   |
| Random-prior  |   0.0624   |   0.7198   |   0.5526   |   0.0788   |   0.7163   |   0.4879   |   0.0641   |   0.7049   |   0.4999   |
| Topk-prior    |   0.1559   |   0.7407   |   0.5681   |   0.1467   |   0.7554   |   0.5227   |   0.1266   |   0.7137   |   0.5033   |
| Freq-prior    |   0.1078   |   0.7311   |   0.5612   |   0.1003   |   0.7351   |   0.5033   |   0.0951   |   0.7190   |   0.4916   |
| BrainMosaic | **0.7506** | **0.8586** | **0.6982** | **0.7144** | **0.8453** | **0.6094** | **0.5520** | **0.8198** | **0.5956** |


### 2. Scalability & Finite Semantic Space

Suggested by Reviewer y3DJ and xT72, we clarify that vocabulary expansion does not lead to performance degradation, due to the intrinsic properties of the continuous semantic space.

**Semantic Density and Coverage**
The continuous embedding space is structurally dense. The core vocabulary already establishes the primary topological distribution of this space. As the vocabulary expands, new words (e.g., synonyms, derivatives) naturally map into existing high-density regions rather than creating distinct new conceptual dimensions. Consequently, the effective "semantic volume" the model needs to decode remains stable and bounded.

**Cognitive Saturation: Theoretically, the number of distinguishable semantic concepts in cognition is finite.**

Theoretically, the number of distinguishable semantic concepts in cognition is finite. Research indicates that approximately 12,000+ cue words are sufficient to cover both the core and long-tail regions of the human semantic space [1]. Our experimental configuration (30,000 units) already exceeds this threshold, ensuring that the semantic space is saturated and robust.

---

> ### Author Response · Authors · 2025-11-21
> **Global Comment (2/3)**
>
> ### 3. Metrics Validation & Qualitative Analysis
>
> Inspired by Reviewer y3DJ, We provide a unified evaluation bridging quantitative benchmarks and qualitative interpretability (Appendix D).
>
> **1. Qualitative Analysis:**
> While we include standard metrics for completeness, our qualitative analysis highlights why they are often ill-suited for Semantic Intention Decoding. As illustrated in Figure 2(b) (Appendix D), traditional n-gram metrics fail to capture true semantic recovery due to four common linguistic phenomena:
>
> - Polysemy: A character match (e.g., "好" for very) may be scored highly even if the context implies a different meaning (e.g., "好" for good).
> - Synonymy: Valid paraphrasing (e.g., "playing on phone" vs. "scrolling on phone") is penalized by rigid token matching, despite preserving the core intent.
> - Keyword Proximity: Sentences that share keywords but lack semantic relevance often receive inflated BLEU scores.
> - Broken Syntax: Concatenations of keywords without logical structure can score well on n-gram precision but fail on our proposed SRS metric, which requires semantic coherence.
>
>
> **2. Qualitative Visualization:**
>
> - **Concept-Level**:  Analysis confirms our embedding-based metrics effectively distinguish between true synonyms (similarity>0.8) and unrelated words, validating our threshold choices.
> - **Intent-Level Examples**: We provide examples fpr side-by-side comparisons of Ground Truth, Model Predictions, and Baselines. These examples visually confirm that our model recovers the intent of the thought.
>
> **3. Standard Metrics Supplement:**
>
> - **BERTScore-F1**: Adopted as a primary metric in main experiments. BrainMosaic demonstrates consistent superiority, confirming its advantage in semantic alignment.
> - **BLEU/WER**: Although we highlight the limitations of these metrics for Semantic Intention Decoding, we report the full results to serve as a comprehensive reference in Appendix D.
>
> [1] De Deyne, Simon, et al. "The “Small World of Words” English word association norms for over 12,000 cue words."

---

### Author Response · Authors · 2025-11-21
**Global Comment (3/3): Summary**

We sincerely thank all the reviewers for your time and the constructive feedback provided.
We found the comments extremely insightful and have used them to significantly strengthen the rigor and clarity of our work.

In response to your suggestions, we conduct a comprehensive revision of the manuscript.
The key improvements are summarized below:

- **Enhanced Evaluation Rigor**: We update and reorganize the main table to include BERTScore-F1. Additionally, we incorporate standard deviations and statistical significance tests in Table 3, and standardize the text embedding configuration using 'Doubao' model to ensure a unified semantic space for more intuitive comparisons. The sensitivity analysis of other text embedding models is decoupled and detailed in Appendix E.
- **Systematic Component Ablation**: We introduce a rigorous ablation study in Section 3.5, explicitly isolating and quantifying the distinct contribution of each core component within the BrainMosaic.
- **Qualitative Analysis & Metric Validation**: We provide intuitive examples of output sentences in Appendix D, qualitatively demonstrating how our proposed metrics better capture semantic recovery.
- **Extensive Supplementary Experiments**: We perform a series of new experiments inspired by reviewers:
    - Appendix C: Random/Text-Prior Baselines
    - Appendix D: Supplement of standard surface metrics (BLEU/WER)
    - Appendix E: Analysis of Embedding Models.
    - Appendix G: Hyper-parameter analysis for the number of semantic slots.
    - Appendix I: Regional Contribution Analysis.
- **Clarifications**: We refine the manuscript to address specific details, including:
    - Adding missing citations (line 78) and theoretical explanations for Scalability (line 218).
    - Formalizing the definition of $MUS_{exp}$ (line 345).
    - Clarifying the data splitting protocol, repeated experiments (line 359).
    - Fixing typo in dataset name (line 394).

We hope these revisions comprehensively address the concerns raised and firmly establish the validity and contribution of our work. A detailed point-by-point response follows.

---

### Author Response · Authors · 2025-12-02
**Final Summary of Revisions and Responses**

Dear AC, SAC, PC and Reviewers,

We sincerely thank all reviewers for their time and constructive feedback.
We appreciate the positive assessments regarding the **novelty of the SID framework** (MrKn, xT72, y3DJ), the **modular and interpretable design** (MrKn, y3DJ), the **comprehensive empirical validation** (MrKn, y3DJ), the **overall contribution and presentation** (MrKn, xT72, y3DJ), and the **theoretical grounding** of the approach (y3DJ).

We are also grateful for the reviewers' insightful suggestions. We have carefully addressed all comments and incorporated all feasible improvements. Below we summarize the key revisions in the updated manuscript:

- We add **ablation studies and component analyses**:
    - Following reviewer MrKn, we add a systematic ablation study isolating and quantifying the contribution of each core component within the BrainMosaic (Section 3.5).
    - Following reviewer MrKn and xT72, we decouple the contribution of the EEG decoding module and show that the performance gains stem from accurate EEG decoding rather than LLM priors or text biases (Appendix C).
    - Following reviewer y3DJ, we provide a sensitivity analysis of the text-embedding module, showing that embedding quality is a key determinant of decoding performance. This also implies a promising property for our framework: as LLM embedding technologies continue to advance in expressiveness and granularity, our method will naturally inherit these improvements, potentially leading to further gains in decoding accuracy without architectural changes (Appendix E).
- We add more **comprehensive and intuitive experiments and comparisons**:
    - Following reviewer y3DJ, we provide qualitative examples jointly reflecting surface-level and semantic-level metrics to illustrate how our proposed metrics better capture semantic recovery (Appendix D).
    - Following reviewer y3DJ and xT72, we adopt BERTScore-F1 as a primary evaluation metric, update the main results accordingly (Tables 3-4), and include traditional surface-level metrics (BLEU-1/2/4, WER) for completeness (Appendix D).
    - Following reviewer y3DJ, we add an analysis and visualization of brain-region contributions, consistent with existing neuroscience findings (Appendix I).
- We strengthen the **experimental rigor** through statistics and additional analyses:
    - Following reviewer MrKn and y3DJ, we clarify that all experiments are conducted under an in-subject setting and explain the rationale (lines 359, 831).
    - Following reviewer MrKn and y3DJ, we add statistical checks confirming that the model does not exploit sentence length or contextual artifacts, eliminating risks of unintended data leakage.
    - Following reviewer y3DJ, we incorporate standard deviations and statistical significance tests, showing that BrainMosaic achieves statistically significant improvements (p < 0.001) over the best baselines across all metrics and datasets (Table 3).
- We further demonstrate the **robustness** of the proposed BrainMosaic:
    - Following reviewer xT72, we analyze robustness to the choice of the number of semantic slots $K$, showing stable performance beyond an interpretable threshold (Appendix G).
    - Following reviewer xT72, we clarify the nature of intrinsic EEG noise with supporting neuroscience literature and add synthetic-noise experiments demonstrating robustness under varied noise conditions.
- Following reviewer y3DJ and xT72, we clarify the **scalability** of our framework, showing that vocabulary expansion does not degrade performance due to properties of the continuous semantic space, and we add relevant theoretical references (lines 218-221).
- Following reviewer MrKn, we add missing citations (line 78), formalize the definition of $MUS_{exp}$ (lines 344-348), and correct the dataset name typo (line 394).

We reiterate our sincere gratitude to the reviewers for their insightful comments and to the AC for overseeing this process.
We believe the current revision has substantially improved the manuscript by incorporating these valuable insights. We hope the updated version and our responses serve as a solid basis for the AC's final recommendation.
We deeply appreciate your efforts in helping us improve our work.

Best Regards,

The Authors

---

### Meta-Review · Area_Chair_aeJ8 · 2025-12-21

**Summary:**

The paper introduces Semantic Intent Decoding and the BrainMosaic architecture to translate EEG and SEEG signals into natural language. By modeling intent as a flexible, compositional set of semantic units, the framework avoids the constraints of fixed-class classification and unconstrained generation. The system utilizes a semantic decomposer, a retriever aligned with text embeddings, and an LLM for guided reconstruction. Evaluations across diverse multilingual datasets demonstrate improved interpretability and communication effectiveness.

**Strengths:**
1. The proposed framework offers a novel perspective by treating brain-to-text decoding as a set of unordered semantic units rather than traditional sequential output. This approach is well-motivated by neuroscience evidence and offers a more plausible representation of communicative intent.
2. BrainMosaic features a well-structured and interpretable pipeline where design principles map directly to specific neural and linguistic modules. The transition from query slots to ranked retrieval and prompted generation provides a more transparent alternative to standard black-box models.
3. The experimental design is robust, testing the framework across multiple languages and different signal modalities such as EEG and SEEG. This comprehensive evaluation across several public and private datasets demonstrates the generalizability and consistency of the proposed approach.

**Weaknesses:**
1. The evaluation lacks standard surface-level metrics like WER or BLEU, and there is an absence of random baselines to verify how much information is truly derived from neural signals. Without these comparisons, it is difficult to benchmark the model against state-of-the-art methods or rule out the influence of language model priors.
2. Several methodological details regarding the data split and potential leakage between training and testing sets are underspecified.
3. The paper lacks qualitative examples of reconstructed sentences and robustness tests against input noise, which are essential for validating the practical utility of the system.

**Reviewer Concerns:**

During the rebuttal, the authors integrated ablation studies and component analyses, provided more comprehensive experiments, strengthened the empirical soundness, and enhanced the robustness evaluation, as suggested by the reviewers. Although follow-up responses from all reviewers are still pending, it is highly likely that some will upgrade their overall assessments.

**Reviewer Scores:**

Although follow-up responses from all reviewers are still pending, it is highly likely that some will upgrade their overall assessments.

---

### Decision · Program_Chairs · 2026-01-26

Accept (Poster)